**RESEARCH**

# Characterization of large-scale genomic differences in the first complete human genome

Xiangyu Yang[1†], Xuankai Wang[1†], Yawen Zou[1], Shilong Zhang[1], Manying Xia[1], Lianting Fu[1], Mitchell R. Vollger[2], Nae-Chyun Chen[3], Dylan J. Taylor[4], William T. Harvey[2], Glennis A. Logsdon[2], Dan Meng[1], Junfeng Shi[5,6], Rajiv C. McCoy[4], Michael C. Schatz[3,4], Weidong Li[1], Evan E. Eichler[2,7], Qing Lu[1] and Yafei Mao[1,6*]

†Xiangyu Yang and Xuankai Wang contributed equally to this work.

*Correspondence:
yafmao@sjtu.edu.cn

[1] Bio-X Institutes, Key Laboratory for the Genetics of Developmental and Neuropsychiatric Disorders, Ministry of Education, Shanghai Jiao Tong University, Shanghai, China
[2] Department of Genome Sciences, University of Washington School of Medicine, Seattle, WA, USA
[3] Department of Computer Science, Johns Hopkins University, Baltimore, MD, USA
[4] Department of Biology, Johns Hopkins University, Baltimore, MD, USA
[5] Shanghai Engineering Research Center of Advanced Dental Technology and Materials, Shanghai, China
[6] Shanghai Key Laboratory of Stomatology, Shanghai Ninth People's Hospital, College of Stomatology, Shanghai Jiao Tong University School of Medicine, Shanghai, China
[7] Howard Hughes Medical Institute, University of Washington, Seattle, WA, USA

## Abstract

**Background:** The first telomere-to-telomere (T2T) human genome assembly (T2T-CHM13) release is a milestone in human genomics. The T2T-CHM13 genome assembly extends our understanding of telomeres, centromeres, segmental duplication, and other complex regions. The current human genome reference (GRCh38) has been widely used in various human genomic studies. However, the large-scale genomic differences between these two important genome assemblies are not characterized in detail yet.

**Results:** Here, in addition to the previously reported "non-syntenic" regions, we find 67 additional large-scale discrepant regions and precisely categorize them into four structural types with a newly developed website tool called SynPlotter. The discrepant regions (~ 21.6 Mbp) excluding telomeric and centromeric regions are highly structurally polymorphic in humans, where the deletions or duplications are likely associated with various human diseases, such as immune and neurodevelopmental disorders. The analyses of a newly identified discrepant region—the *KLRC* gene cluster—show that the depletion of *KLRC2* by a single-deletion event is associated with natural killer cell differentiation in ~ 20% of humans. Meanwhile, the rapid amino acid replacements observed within *KLRC3* are probably a result of natural selection in primate evolution.

**Conclusion:** Our study provides a foundation for understanding the large-scale structural genomic differences between the two crucial human reference genomes, and is thereby important for future human genomics studies.

**Keywords:** Complete human genome (T2T-CHM13), Human reference genome (GRCh38), Large-scale structural variation, Neurological disease, Immune disorder, Discrepant region, *KLRC* genes

## Background

The first draft human genome published two decades ago has contributed enormously to human genomics, medical genomics, evolutionary genomics, and other fields [1, 2]. Given efforts to refine and construct the sequence from Genome

Reference Consortium, the current human reference genome assembly (GRCh38) has been widely used for understanding human diversity, disease-related variants, and human/primate evolution [3]. The GRCh38 genome assembly has been annotated with abundant resources including gene annotation, gene expression, gene regulation, and others [3]. Despite the high quality of the GRCh38 reference, it still has hundreds of gaps and errors [4]. These gaps and errors represented long-standing obstacles to fully understanding human genomics, especially in repetitive regions [4–10]. With advances in long-read sequencing and computation algorithms, the Telomere-to-Telomere (T2T) Consortium has finally achieved the goal of building a gapless and accurate assembly of a human genome [4–9].

The release of the complete genome (T2T-CHM13) provides the first complete sequence view of centromeres, telomeres, tandem repeat arrays, segmental duplications (segdups), and the p-arms of acrocentric chromosomes in the human genome [4–9, 11]. In addition to the big achievement of the genome assembly itself, the T2T Consortium also provided insights into the organization and function of segdups, centromeres, epigenetic features of repeats and genome, and human genetic variation by comparative genomics and population genetics approaches [4–9]. These efforts significantly extend our biological understanding of human genomics and underscore the advantages of using T2T-CHM13 as a reference for genomic analyses [11]. In the previous studies, 238 Mbp of genomic sequences is identified as "non-syntenic" regions between GRCh38 and T2T-CHM13 [4, 9], representing major large-scale genomic differences between these two assemblies.

The large-scale genomic differences are largely concentrated in complex genomic regions, which play an outstanding role in human disease as well as evolutionary adaptation [10, 12]. For example, segdups of *Notch2NL* are associated with brain development in primate evolution, while a rare microdeletion of *Notch2NL* causes microcephaly in humans [13–16]. Therefore, it is necessary for us to comprehensively characterize the structure and function of the large-scale genomic differences between GRCh38 and T2T-CHM13 for future applications (e.g., genotyping, association, and evolutionary studies). Here, we expand on the comparison of "non-syntenic" regions between T2T-CHM13 and GRCh38 in the previous studies [4, 9], identifying additional large-scale genomic differences between these two assemblies applying an array of additional alignment and visual validation tools. We characterize the genomic regions with at least 10 kbp genomic differences between the two assemblies into four types: insertions, deletions, inversions, and structural divergent regions (SDRs), with respect to GRCh38. We then develop an integrated website tool (SynPlotter, https://synplotter.sjtu.edu.cn/) to validate the discrepant regions and characterize the gene model differences in these regions. In addition, we use the ~300 human genomes from the Simons Genome Diversity Project (SGDP) [17] to test whether these discrepant regions are likely copy number (CN) polymorphic in human populations. We also investigate the functional relationship between discrepant regions and human diseases. Finally, we systematically analyze the evolutionary history of one example of a newly identified discrepant region—the *KLRC* gene cluster—in human populations and other non-human primates.

## Results

### Large-scale genomic discrepant regions

More than 570 "non-syntenic" regions (~238Mbp) have been identified between the T2T-CHM13 and GRCh38 genome assembly with a 1Mbp syntenic interval approach [4, 9]. Here, to more completely and precisely characterize the structural types of the large-scale genomic differences between the two genome assemblies, we applied three additional alignment tools (PAV [18, 19], minigraph [20], PBSV (https://github.com/PacificBiosciences/pbsv)) to expand on the non-syntenic regions originally identified by LASTZ [4, 9]. We identified the 694 structural variants (SVs, ≥ 10 kbp) with three independent methods (Additional file 2: Tables S1-S3). Next, we developed an integrated website tool (SynPlotter) that is designed to visualize and cross-validate the syntenic relationship between GRCh38 and T2T-CHM13 by integrating multiple aligners (e.g., minimap2 and numer) and publicly available visualization tools (e.g., dotplot and SafFire (unpublished, https://mrvollger.github.io/SafFire)) [21, 22].

Excluding the SVs in centromere and telomere regions, we validated 238 of 274 large SVs (validation rate: 86.9%) with the advance of our validation tool (Additional file 2: Table S4). Next, we integrated our validated large SVs with the validated "non-syntenic" regions (Fig. 1a, Additional file 1: Fig. S1, Additional file 2: Table S4) to find 590 discrepant genomic regions between GRCh38 and T2T-CHM13 in total (Fig. 1b). Of these, 295

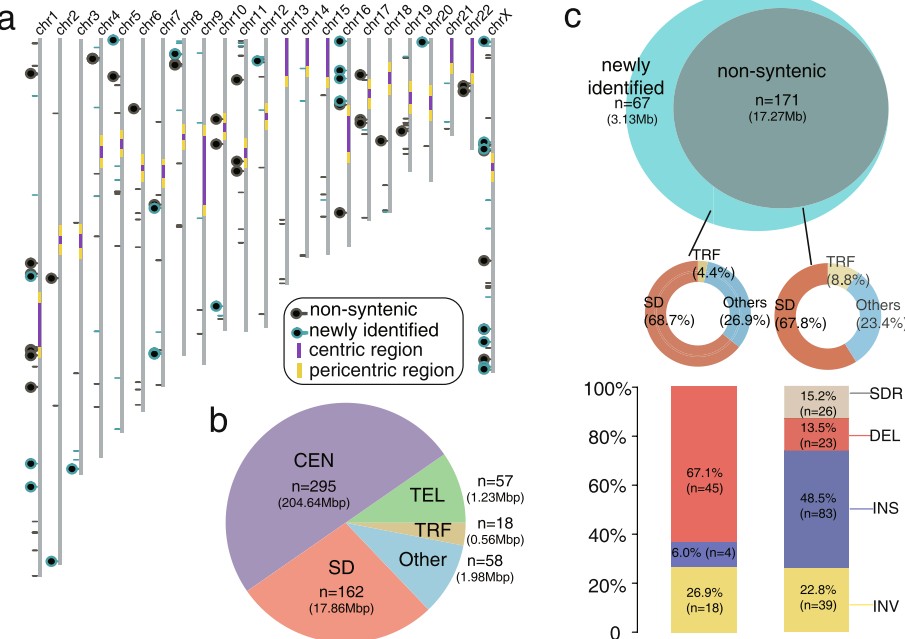

**Fig. 1** The discrepant genomic regions between GRCh38 and T2T-CHM13. **a** Schematic of the T2T-CHM13 assembly depicts the centromere location (purple and yellow), "non-syntenic" region (black lines and circles), and newly identified discrepant region (cyan lines and circles). Regions containing genes are represented with circles. **b** Pie chart of genomic structure annotations of the 590 discrepant regions. The proportion of regions in centromeres (CEN), telomeres (TEL), segdups (SD), tandem repeats (TRF), and others are shown in light purple, green, dark red, yellow, and blue. **c** Venn diagram shows the comparison of the discrepant regions between the previous studies [4, 9] and this study. The genome structure annotations of "non-syntenic" and newly identified regions are shown in the middle panel. The components of structural variant types of "non-syntenic" and newly identified regions are shown in the bottom panel

regions are in centromeres (204.64 Mbp), 57 regions are in (sub)telomeres (1.23 Mbp), 162 regions are in segdups (17.86 Mbp), 18 regions are in tandem repeats (0.56 Mbp), while 58 regions occur in other parts of the genome (1.98 Mbp) (Fig. 1b).

Despite the functional importance of telomeres and centromeres, it is informative to compare centromeres and telomeres between the two genome assemblies in our study because most of them are incorrectly assembled, gaps, or decoys in GRCh38 [4–9, 23]. Therefore, we excluded the 352 discrepant regions in centromeres and telomeres in the following analyses. In addition, instead of using the term "non-syntenic", we refined the characterization of the large-scale discrepant regions by categorizing them into four types (including: insertions, deletions, inversions, and SDRs, with respect to GRCh38) with more precise breakpoints (Fig. 1c). There are 23 deletions (1.51 Mbp), 83 insertions (3.42 Mbp), 39 inversions (10.47 Mbp) and 26 SDRs (1.87 Mbp) in the previously reported "non-syntenic" regions (total: 17.27 Mbp) (Fig. 1c). Relative to the previously reported "non-syntenic" regions, here, we found 67 newly identified discrepant regions, of which the number is ~40% greater than that of the reported "non-syntenic" regions (Fig. 1c). The 67 newly identified regions (total: 3.13 Mbp) include 45 deletions (1.7 Mbp), 4 insertions (0.06 Mbp), and 18 inversions (1.37 Mbp) (Fig. 1c). The number of deletions in the newly identified set is higher than in the "non-syntenic" regions ($p < 0.001$, chi-square test).

### Gene structure difference in the CN polymorphic discrepant regions

Of the 238 discrepant regions, 63 of them include 153 genes, such as *TBC1D3*, *AMY1*, *GPRIN2*, and *NOTCH2NL* (reported in the "non-syntenic" regions) [9]. Of these, 53 protein-coding genes are in the 25 newly identified discrepant regions, including *ZDHHC11B*, *GSTM2*, *CFHR3*, *CFHR1*, *CR1*, and *KLRC2* (Additional file 2: Table S5). We observed the depletion of *GSTM1* in T2T-CHM13 by a ~17 kbp deletion and the *GSTM* is inferred as CN polymorphic in the SGPD human samples (Fig. 2a). The gene models showed that a few amino acids of *GSTM1* are different from that of *GSTM2* (Fig. 2a). In addition, we observed an ~18.5 kbp deletion in T2T-CHM13, resulting in the depletion of eight exons (450 amino acids) in *CR1* (Fig. 2b). We examined the length of *CR1* gene in the 94 long-read human genome assembly from the Human Pangenome Reference Consortium (HPRC) [24–26] and the length of *CR1* in 79 assemblies coincides with that of T2T-CHM13. This suggests that T2T-CHM13 carries the major allele of *CR1* (allele frequency: 0.84) (Additional file 1: Fig. S2). As previous study reported, the depletion of *ZDHHC11B* is found in T2T-CHM13 by a ~98 kbp deletion, with respect to GRCh38 (Additional file 1: Fig. S3) [27]. Here, we also observed that the two exons are deleted in *ZDHHC11B* compared to *ZDHHC11* (Additional file 1: Fig. S3). In addition, another ~85 kbp genomic region, including *CFHR1* and *CFHR3*, is deleted in T2T-CHM13, with respect to GRCh38 (Additional file 1: Fig. S4). In total, we examined the gene model difference in the discrepant regions, 22 and 21 protein-coding genes differ in the deletions and insertions, respectively (Additional file 1: Fig. S5, Additional file 2: Table S6).

We assessed whether the discrepant regions are likely CN polymorphic in the human genome. We used the standard deviation (s.d.) of the CN as an index to represent the level of polymorphisms (see "Methods") [28]. To reduce the CN estimation bias, we excluded

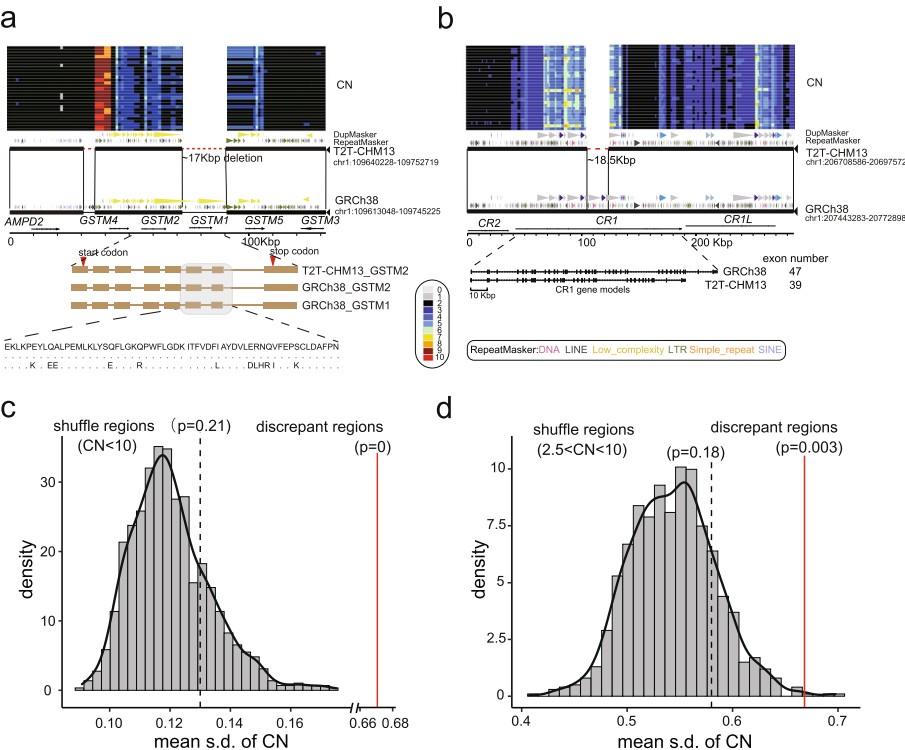

**Fig. 2** Gene structure differences in the discrepant regions. **a** The depletion of GSTM1 in the T2T-CHM13 genome assembly by a ~ 17 kbp deletion. The CN heatmap inferred from SGPD is shown in the top panel. The miropeat synteny relationship shows structural variation with repeat, duplication, and gene annotation. The exon schematic with amino acid alignment shows the gene-model difference in the two assemblies. **b** The depletion of eight exons of *CR1* in T2T-CHM13 by ~ 18.5 kbp deletion. **c** The distribution of the mean of s.d. of CN shows the mean s.d. of 131 discrepant regions (mean = 0.735, red line) is significantly higher than the simulated null distribution of s.d. of CN (CN < 10, empirical $p$ = 0). The black line represents the observed mean of s.d. of CN of the regions where the CN is less than 10. **d** The distribution of the mean s.d. of CN shows the mean s.d. of 131 discrepant regions (mean = 0.735, red line) is significantly higher than the simulated null distribution of s.d. of CN (2.5 < CN < 10, empirical $p$ = 0). The black line represents the observed mean s.d. of CN of the regions where the CN is less than 10 and greater than 2.5

the regions where the CN is greater than 10 in the following analyses. We observed that the mean s.d. of the CN of the 131 discrepant regions (mean = 0.67) is ~ 5-fold greater than that of the whole genomic regions (mean = 0.13, empirical $p$ = 0) (Fig. 2c and Additional file 1: Fig. S6), as expected. We next tested whether the discrepant regions are more likely CN polymorphic than the CN variable regions (CN > 2.5 and CN < 10). We observed the mean s.d. of the CN of the 131 discrepant regions (mean = 0.58) is ~ 1.2-fold greater than that of the CN variable regions (mean = 0.58, empirical $p$ = 0.003) (Fig. 2d). Yet, we did not observe a significant difference between the median s.d. of the CN of the 131 discrepant regions (median = 0.46) and that of the CN variable genomic regions (median = 0.4, empirical $p$ = 0.07) (Additional file 1: Fig. S6). The simulation tests imply that the discrepant regions are more likely CN polymorphic than the genome-wide average, maybe even than the CN variable regions in the human genome. We also used the long-read human pangenome graph (HPRC and HGSVC, $n$ = 152) to genotype the 87 insertions and 68 deletions [18, 26]. We found that 105 SVs (66 deletions and 39 insertions) can be genotyped in the pangenome

graph (105/155, 67.74%) and the results suggest that the discrepant regions are polymorphic in humans (Additional file 2: Table S7).

### Disease-related loci are associated with the large-scale discrepant regions

We integrated the reported morbid copy number variants (CNVs) and genomic disorder CNVs that associated with more than 50 disease phenotypes, including neurodevelopmental disorders, abnormality of the immune system, and others [29–31]. We next queried whether the discrepant regions are more likely associated with the reported disease-related CNVs. With genome-wide permutation analysis (see Methods), we found that the discrepant regions are significantly co-localized with disease-relevant CNVs (empirical $p = 0.003, \sim 1.7$-fold excess) (Additional file 1: Fig. S7). To better characterize the genes/genomic coordinates relevant to disease, we surveyed the literature and DECIPHER database for the aforementioned discrepant regions and found 27 discrepant regions associated with human diseases; 18 of them are newly identified discrepant regions (Table 1).

The genes in the 27 disease-related discrepant regions are enriched in the neuroblastoma breakpoint family domain ($p = 3.9e-5$), complement and coagulation cascades ($p = 7.5e-4$), glutathione metabolic process ($p = 4.4e-3$), and antimicrobial ($p = 5.5e-5$) by the gene ontology (GO) enrichment analysis (Additional file 2: Table S8). Therefore, the rare microdeletions or microduplications of these discrepant regions mainly affect the development and function of the circulatory system (urinary system disease (e.g., chromosome 1p13.3)), immune system (COVID-19 (e.g., 6p21.32, 12p13.2)), and nervous system (bipolar disorder/schizophrenia [42] (e.g., 10q11.22) and autism spectrum disorder (e.g., 16p12)) (Table 1). We also found some genes within the discrepant regions that are proven to be functionally well-known and pathogenic. For example, *KLRC2*, located in a newly identified discrepant regions, is involved in immune cell maturation and subtype differentiation [45]. The KLRC2 protein (also: NKG2C) can bind to CD94 and HLA-E to form a functional complex [53], and thus, the depletion of *KLRC2* is likely to have a significant impact on the development of severe COVID-19 [44]. In the visual cortex, microglial CD94/KLRC2 is necessary for regulating the magnitude of ocular dominance plasticity during the critical period of development [54]. *GSTM1* (Glutathione S-Transferase Mu 1) encodes a member of a superfamily of antioxidant enzymes, which is important in kidney disease progression [32]. *ZDHHC11B* (Zinc Finger DHHC-Type Containing 11B) is involved in a network that promotes the proliferation of Burkitt lymphoma cells [38]. *CFHR3* and *CFHR1*, belonging to Complement factor H (CFH), plays an essential role in regulating the alternative pathway of the complement system [35]. These results suggest that the discrepant genomic regions are functionally important.

### The diversity of *KLRC2* characterized with the 94 long-read and 2,504 short-read human genomes

We observed that *KLRC2* is deleted by a 15.4 kbp deletion variant in T2T-CHM13, with respect to GRCh38 (Fig. 3a). This discrepant region is CN polymorphic in human populations as evidenced by SGPD read-depth genotyping [55–57] (Additional file 1: Fig. S8). To better characterize the diversity of the *KLRC* region, we systematically investigated the discrepant region with the 94 long-read genome assemblies from the HPRC dataset [24–26]. We found 1 duplication and 11 deletions of *KLRC2* in the 94 long-read genome

**Table 1** The discrepant regions associated with human diseases

| Cytobands | CHM13_Position | Hg38_Position | Type | Reported CNV | Genes | Disease |
|---|---|---|---|---|---|---|
| 1p13.3 | chr1:109711485–109711489 | chr1:109682999–109701443 | DEL | | *GSTM1* | Urinary system disease [32] |
| 1p21.1 | chr1:103546781–103735057 | chr1:103697900–103697950 | INV | | *AMY1A, AMY1B, AMYP1* | Neurological disease [33] |
| 1p36.13 | chr1:16007445–16027869 | chr1:16565700–16565800 | INS | | *NBPF1* | Neurodevelopmental disorders [34], Cancer |
| 1q21.1-1q21.2 | chr1: 143959965–143983984 | chr1:146251047–148716074 | INV | | *BCL9, NOTCH2NLB, CHD1L, NBPF12, PRKAB2, FMO5, ACP6, GJA8, GPR89B, NBPF11, NBPF14, PPIAL4G,* | Neurodevelopmental disorders [13–16] |
| 1q31.3 | chr1:196105143–196105148 | chr1:196758727–196843410 | DEL | | *CFHR3, CFHR1* | Immunological disease [35] |
| 1q32.2 | chr1:206810072–206810076 | chr1:207542838–207561393 | DEL | | *CR1* | Neurological disease [36, 37] |
| 2q13 | chr2:110517534–110698558 | chr2:110095177–110276210 | INV | Morbid CNV & Disease-related CNV | *NPHP1, MALL, MTLN* | Neurodevelopmental disorders [29, 30],Neurological disease [31] |
| 3q29 | chr3:198347865–198715835 | chr3:195641035–195995576 | DEL | Disease-related CNV | *MUC20, MUC4, TNK2* | Neurodevelopmental disorders [29, 30] |
| 5p15.33 | chr5:684792–685093 | chr5:686991–779053 | DEL | | *ZDHHC11B* | Cancer [38] |
| 6p21.32 | chr6:32339743–32356931 | chr6:32486765–32530206 | DEL | | *HLA-DRB5* | Immunological disease [39] |
| 6q26 | chr6:161865491–161959834 | chr6:160612509–160612509 | INS | | *LPA* | Cardiovascular disease [40] |
| 7q35 | chr7:145,477,647–145477649 | chr7:144197172–144295737 | DEL | | *OR2A42, OR2A7, CTAGE8* | Cancer [41] |
| 8p23.1 | chr8:750030–11722000 | chr8:8022351–12234558 | INV | Morbid CNV | *DLGAP2, MYOM2, CLN8, ARHGEF, CSMD1, MCPH1, ANGPT2, PRR23D1, DEFB103B, DEFB103A, DEF104A, DEF105A, XKR6, SOX7, TNKS, PPP1R3B, PPAG1, CTSB, ANGPT2, AGPAT5, ERI1, MSRA, DEFA5, FDFT1, GATA4, MFHAS1, PRSS5* | Developmental disorders [29, 30] |

**Table 1** (continued)

| Cytobands | CHM13_Position | Hg38_Position | Type | Reported CNV | Genes | Disease |
|---|---|---|---|---|---|---|
| 10q11.22 | chr10:48671598–48719249 | chr10:47780140–47870155 | SDR | Disease-related CNV | *GPRIN2* | Neurological disease [42] |
| 11p15.5 | chr11:1076897–1087865 | chr11:1017980–1017990 | INS | | *MUC6* | Neurological disease [43] |
| 12p13.2 | chr12:10315804–10315827 | chr12:10429009–10444430 | DEL | | *KLRC2* | COVID-19 [44], Immunological disease [45] |
| 16p11.2 | chr16:30492288–30594258 | chr16:30207700–30207750 | INS | Disease-related CNV | *NPIPB13, BOLA2B* | Neurodevelopmental Disorders [29, 30, 46] |
| 16p12.1–12.2 | chr16:28619710–29091966 | chr16:28339205–28811381 | INV | Disease-related CNV | *SULT1A1, SULT1A2, NPIPB8, NPIPB6 EIF3CL, NPIPB7, CLN3, IL27, EIF3C, NPIPB9* | Neurodevelopmental disorders [29, 30], Neurological disease [31] |
| 17p11.2 | chr17:16716173–16767175 | chr17:16813513–16821452 | SDR | Disease-related CNV | *LGALS9C* | Neurodevelopmental disorders [29, 30] |
| 17q12 | chr17:37341285–37441106 | chr17:36393230–36459266 | SDR | Disease-related CNV | *CCL3L1, CCL4L2, TBC1D3F* | Neurodevelopmental disorders [29, 30] |
| 19q13.2 | chr19:42710594–42726422 | chr19:39906200–39906250 | INS | | *FCGBP* | Reproductive system disease [47] |
| 20p13 | chr20:1629529–1629530 | chr20:1580346–1613395 | DEL | | *SIRPB1* | Immunological disease [48] |
| 22q11.23 | chr22:24380000–24462473 | chr22:23932712–24000827 | SDR | Disease-related CNV | *GSTT2, GSTT4, DDT* | Neurodevelopmental disorders [29, 30], Neurological disease [31] |
| Xp11.22 | chrX:50939534–50996879 | chrX:51668108–51725222 | INV | | *CENPVL1, CENPVL2* | Neurodevelopmental disorders [49] |
| Xp22.33 | chrX:1307333–1307498 | chrX:1465426–1506104 | DEL | | *P2RY8* | Immunological disease [50] |
| Xq26.3 | chrX:134047172–134104452 | chrX:135721633–135795043 | SDR | | *CT45A1, CT45A3, CT45A5,* | Cancer [51] |
| Xq28 | chrX:147946883–147987904 | chrX:149681127–149722143 | SDR | | *MAGEA11* | Cancer [52] |

Morbid CNV refers to Ref. [29, 30]

Disease-related CNV refers to the Ref. [31]

assemblies (Additional file 2: Table S9). We refined the breakpoints of the duplication and deletion of *KLCR2* in the T2T-CHM13 and HG002_hap [19] genome assemblies at single-base pair resolution to understand the mechanisms of the structural variation.

There are four *KLRC* genes in the *KLRC* discrepant region, wherein a segdup including *KLRC2* and *KLRC3* is configured in a direct orientation [58] (Fig. 3). The configuration provides a genetic basis for microdeletions and microduplications of *KLRC2*. The syntenic relationship of the *KLRC* gene cluster between T2T-CHM13 and GRCh38 revealed that the left breakpoint of the *KLRC2* deletion is located within *AluYm1*, and the right

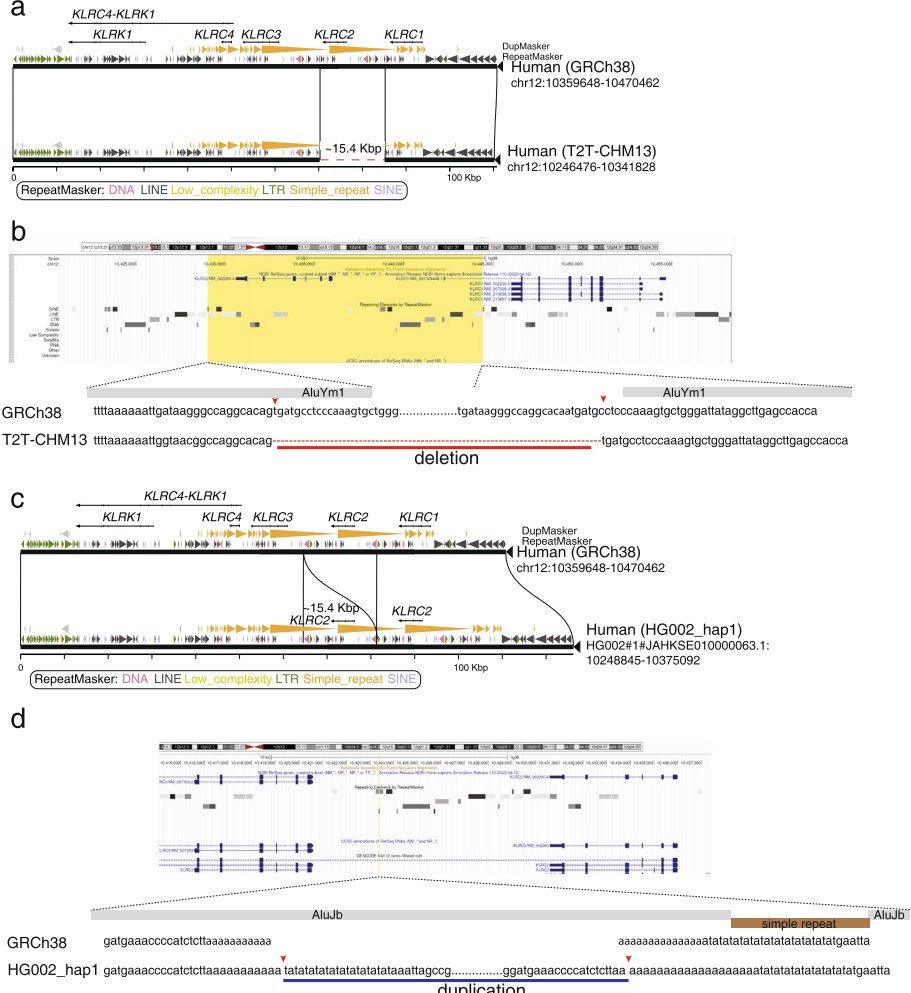

**Fig. 3** The syntenic comparison between different *KLRC* gene cluster haplotypes. **a** A ~ 15.4 kbp deletion in the T2T-CHM13 genome assembly results in the complete loss of *KLRC2*. Gene structure, duplication, and repeat annotations are shown in the miropeat diagram. **b** A screenshot of the *KLRC2* region from the UCSC Genome Browser is shown in the top panel. The yellow box represents the ~ 15.4 kbp deleted sequence in the T2T-CHM13 genome assembly. The nuclear sequencing alignment of the breakpoints is shown in the bottom panel. Two *Alu* elements surrounding the breakpoints are shown in gray bars. **c** A ~ 15.4 kbp duplication in the HG002_hap1 genome assembly results in the complete duplication of *KLRC2*. Gene structure, duplication, and repeat annotations are shown in the miropeats diagram. **d** A screenshot of the *KLRC2* region from the UCSC Genome Browser is shown in the top panel. The yellow line represents the position where the 15.4 kbp duplicated sequence is in the HG002_hap1 genome assembly. The nuclear sequencing alignment of the breakpoints is shown in the bottom panel. The disrupted *Alu* element within the breakpoints are shown in gray bars and a simple repeat disrupting the *Alu* element is shown in a brown bar

breakpoint of the *KLRC2* deletion is 3 bp away from another *ALuYm1* (Fig. 3b). We also observed a 43 bp repeat motif (tgatgcctcccaaagtgctgggattataggcttgagccacca) at both breakpoints (Fig. 3b). In addition, we refined the breakpoints of the *KLRC2* duplication in the HG002_hap1 assembly (Fig. 3c). We found that the duplication sequences (~ 15.4 kbp) are inserted in an *AluJb* element and the *AluJb* element is disrupted by a simple repeat insertion in GRCh38 (Fig. 3d). The breakpoints are located within ploy adenine (polyA) sequences in GRCh38 (Fig. 3d).

Our analysis of the long-read HPRC haplotypes ($n = 94$) identified three haplotypes of the *KLRC* gene cluster, including 0, 1, and 2 copies of *KLRC2*, respectively. Next, we used the SUNK (singly unique nucleotide *k*-mer) mapping and read-depth genotyping approaches [55–57] to infer the three haplotypes in 2504 human genomes from the 1000 Genome Project (1 KG) [59]. We found that 19, 78, and 3% of modern humans contain 0, 1, and 2 copies of *KLRC2*, respectively (Fig. 4a, Additional file 2: Table S10). The haplotype with a depleted *KLRC2* ("*KLRC*-hap2") occurs more frequently in African (e.g., Esans: 25.45%) and East Asian (e.g., Kinhs: 26.24%) populations but is observed less frequently in American (e.g., Peruvians: 3.8%) and South Asian (e.g., Pakistans: 9.71%) populations (Additional file 2: Table S11).

To study whether the depletion of *KLRC2* is a recurrent or a single-deletion event in human populations, we used the ~12.7 kbp *KLRC* gene cluster genomic regions, including both *KLRC2* and *KLRC3*, to reconstruct the phylogenetic tree of the 94 long-read human samples. The results showed that the *KLRC2* depletion haplotypes from different human groups are monophyletic (Fig. 4b), suggesting that the *KLRC*-hap2 (*KLRC2* depletion) deletion arose once in human population history.

### Gene expression and NK cell differentiation between two KLRC haplotypes

To investigate potential functional effects of different *KLRC2* haplotypes in humans, we identified six single-nucleotide variants (SNVs) that distinguish *KLRC*-hap2 (*KLRC2* depletion) from *KLRC*-hap1 (one copy *KLRC2*) with the 94 long-read genome assemblies. We examined the linkage disequilibrium (LD) of the *KLRC* gene cluster among 2504 high-coverage genomes from the 1 KG human population. In general, the *KLRC* gene cluster shows significant LD (LD: $r^2 > 0.5$, D' $> 0.5$) (Fig. 4c). In particular, the six SNVs identified in the 94 long-read genome assemblies are tightly linked (LD: $r^2 > 0.9$; D' $> 0.99$) (Fig. 4c). These six SNVs can, thus, be used to infer the deletion haplotype. Indeed, we find that the allele frequencies of the six SNVs of the *KLRC*-hap2 in ~135,000 humans from the gnomAD database are from 19.9 to 20.6%, which coincide with the CN frequency of *KLRC*-hap2 in humans from our above *KLRC* haplotype inference analysis (Fig. 4c and Additional file 2: Table S12).

(See figure on next page.)

**Fig. 4** The structural and functional diversity of *KLRC2* in humans. **a** The proportion of three *KLRC* haplotypes is shown in a pie chart. The *KLRC*_hap1 represents one copy of *KLCR2* shown in purple. The *KLRC*_hap2 represents zero copies of *KLCR2* shown in orange. The *KLRC*_hap3 represents two copies of *KLCR2* shown in dark blue. Distributions of *KLRC*_hap1 (purple), *KLRC*_hap2 (orange), and *KLRC*_hap3 (blue) inferred from the 1 KG human population data across the world are shown on the right panel. **b** The phylogenetic tree of the *KLRC* haplotype genomic regions shows *KLRC*-hap2 (*KLRC2* depletion) is a result of a single-deletion event. The red and blue rectangles show *KLRC*-hap2 and *KLRC*-hap3, respectively. The rest of the humans belong to *KLRC*-hap1. The super population of each human is listed with five color dots. **c** The genomic region (chr12:10,000,000–10,700,000 in GRCh38) with assembled BAC clone, gene, segdup, and LD (D') annotation shows that the *KLRC* gene cluster is probably linked together. The six SNVs distinguished between *KLRC*-hap2 and *KLRC*-hap1 are represented in the cyan lines with their SNP ID. The heatmaps of LD indexes (R2 and D') show that the six SNVs are highly linked in humans. **d** Consistent patterns of associations between the six SNVs and expression levels of *KLRC2* in 35 tissues are shown in the multi-tissue eQTL plots. The positive normalized effect size (NES) values represent the effect of the higher expression on the alternative allele (purple) relative to the reference allele (red). The (unadjusted) *p*-values of the eQTL association are shown in the size variable dots. The allele frequencies of the six SNVs in the gnomAD database are shown on the right. **e** The PheWAS plots for three SNVs (rs1382264, rs2617151, and rs2733845) are significantly associated with immune domain differentiation across GWAS in the GWASALAS database. The particular traits (NKearly:%335+314- and NKeff:%314-R7-) are marked with significant signals [45]

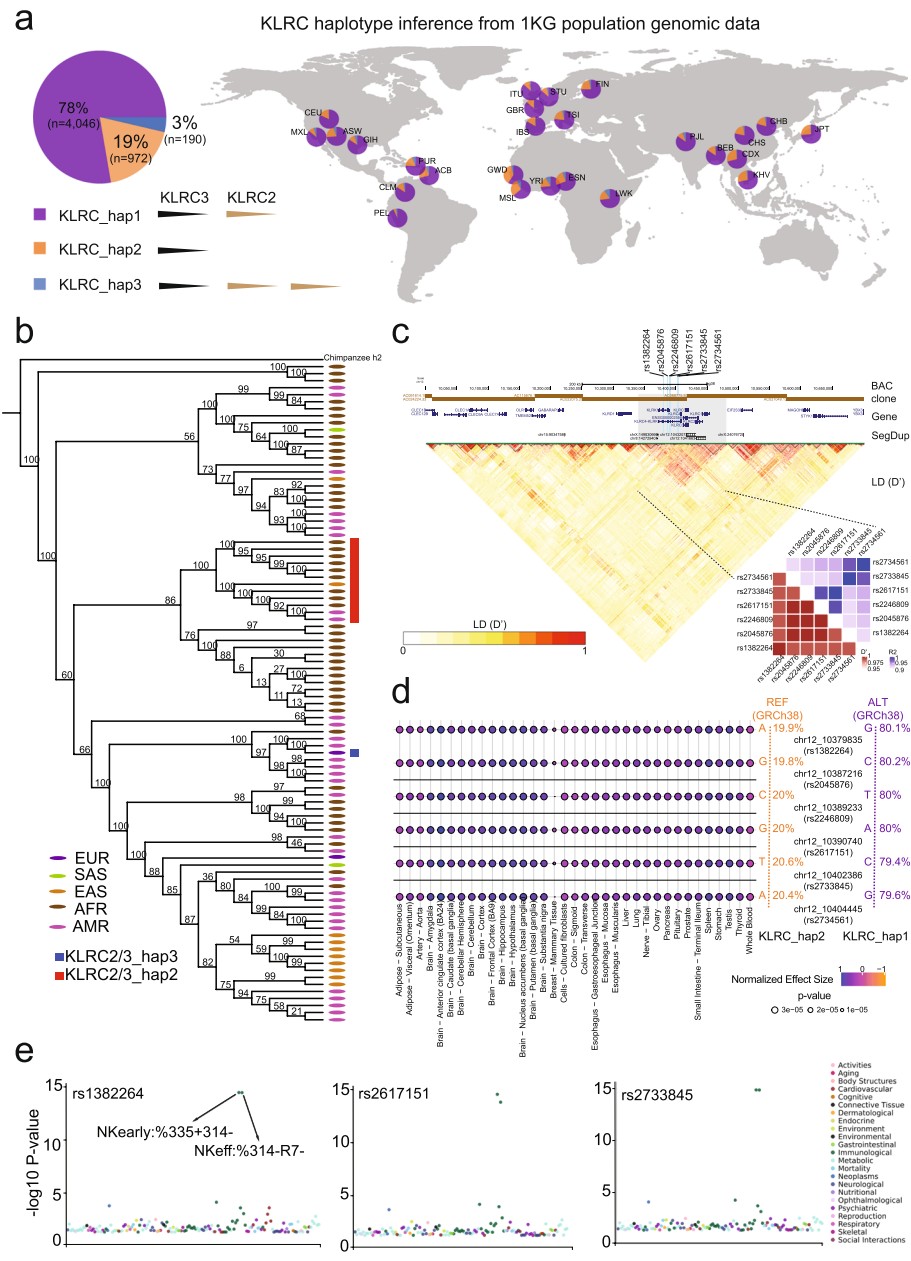

**Fig. 4** (See legend on previous page.)

Of note, the six SNVs are identical between GRCh38 and T2T-CHM13, although apparently distinguish two distinct *KLRC* haplotypes (GRCh38: *KLRC*-hap1, T2T-CHM13: *KLRC*-hap2). We investigated whether this apparent discrepancy could have resulted from a "mixed" haplotype in GRCh38. In GRCh38, we note that the region was assembled by the two distinct bacterial artificial chromosome (BAC) clones (AC022075.29 and AC068775.52) from one donor (RP11) (Fig. 4c). Previous studies have shown that haplotype swaps are usually associated with the overlap boundary of the two adjacent BAC clones [6]. In support of this, our LD analysis within the *KLRC* locus (see "Methods") shows that GRCh38 possesses combinations of alleles that are either in strong positive or strong negative LD, whereas the corresponding region of T2T-CHM13

largely exhibits alleles only in positive LD. Thus, T2T-CHM13 better reflects the haplotype structure of living human populations. LD at the *KLRC* locus extends much further than the randomly selected control locus, which exhibits multiple, shorter haplotype blocks, potentially reflecting differences in the history of recombination within the regions or a deep coalescent deletion (Additional file 1: Fig. S9-10). Taken together, we consider the *KLRC* gene cluster organization in GRCh38 to represent the product of a misassembly of two different haplotypes.

Using the GTEx (Genotype-Tissue Expression) multi-tissue eQTL (expression quantitative trait loci) database (release v8, https://gtexportal.org/), we investigated how these six SNVs relate to *KLRC2* gene expression differences among 54 different tissues. We show that *KLRC*-hap2 SNVs correspond to reduced expression of *KLRC2* gene in 35 tissues (Fig. 4d). In particular, the brain and spleen tissues show the most significant gene expression difference between two haplotypes ($p < 2e - 5$). Further, we investigated the association between these six SNVs with more than 600 phenotypes/traits (GWAS Atlas) and find that three out of six SNVs are significantly associated with immune domain function involving in the NK cell differentiation ($p < 1e - 15$) [45, 60] (Fig. 4e). These results suggest that the depletion of *KLRC2* likely plays a role in the immune differentiation.

### The evolutionary history of *KLRC2* and *KLRC3* in primates

Using sequence read-depth, we investigated CN of *KLRC* genes among a population of non-human primates (NHPs). Our analysis revealed that *KLRC2* and *KLRC3* are also CN polymorphic and that *KLRC2* and *KLRC3* CN in the African great apes is greater than other primates (Additional file 1: Fig. S11). We also investigated the organization of the *KLCR* gene cluster within 16 long-read genome assemblies. The analysis confirmed *KLRC2* and *KLRC3* are CNV in NHPs with a deletion of *KLRC2* and *KLRC3* in two gibbon genome assemblies (Additional file 1: Fig. S12) and three copies of *KLRC2* in two macaque genome assemblies (Additional file 1: Fig. S13). We reconstructed the phylogenic tree of *KLRC2* and *KLRC3* using ~5.5 kbp genomic region. The phylogenetic tree shows that *KLRC2* and *KLRC3* were duplicated within the common ancestor of apes and Old-World monkey at ~19.8 million years ago (mya) (95% CI 10.85-28.97 mya) (Fig. 5a and Additional file 1: Fig. S14). In addition, we found that *KLRC2* is independently duplicated in humans and macaques (Fig. 5a).

We also examined *KLRC2* and *KLCR3* duplicated genes for evidence of natural selection during primate evolution. The pi diversity analysis of the *KLRC* gene cluster based on 94 long-read genome assemblies showed no significant pi diversity drop in the *KLRC2-3* region (chr12:10,299,045–10,307,426) with respect to the entire regions (chr12:10,000,000–10,700,000) (Additional file 1: Fig. S15). Furthermore, considering a large human population dataset (human 1 KG data) and recombination rate, previous selection scans do not detect any positive selection on the *KLRC2-KLRC3* region in human populations neither [61, 62]. In contrast, a branch model estimate of amino acid selection, as defined by PAML and aBSREL (an adaptive branch-site REL test for episodic diversification), found evidence of selection within the *KLRC3* clade ($p = 0.03$, likelihood ratio test) [63, 64] (Additional file 1: Fig. S16, Additional file 2: Table S13). In particular, we identified three amino acids (R224, R227, G229) of *KLRC3* predicted to

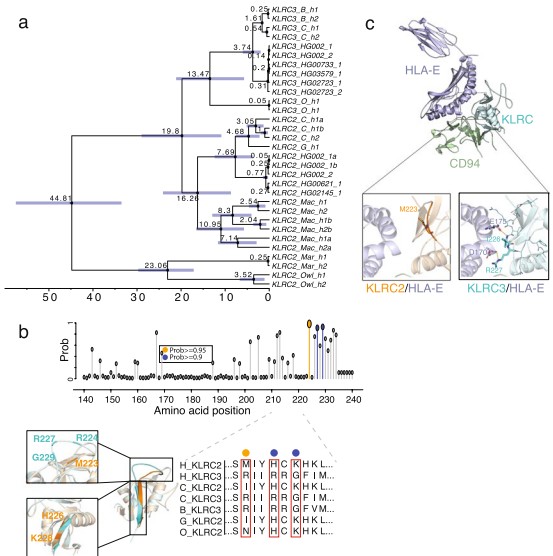

**Fig. 5** The potential functional differentiation between *KLRC2* and *KLRC3* by natural selection in primate evolution. **a** The phylogenetic tree reconstructed from *KLRC2* and *KLRC3* of humans and other NHPs with BEAST2 shows the duplication of *KLRC2* and *KLRC3* occurred at the common ancestor of African great apes. The 95% confidence interval of the estimated age of each node is shown in the blue bar. All nodes are supported by one posterior possibility shown in dark circle dots. The texts: C, B, G, O, Mac, Mar, and Owl in the tips represent chimpanzee, bonobo, gorilla, orangutan, macaque, marmoset, and owl monkey, respectively. **b** The possibility of amino acid under positive selection inferred by the branch-site model in PAML is shown on the top panel. The gray, orange, and blue dots represent the possibility of less than 0.9, between 0.9 and 0.95, or greater than 0.95, respectively. The amino acid alignment of *KLRC2* and *KLRC3* among primates is shown on the bottom right panel. Structure alignment of predicted structures of KLRC2 from residue 118 to 231 (orange, Uniprot: P26717) and KLRC3 from residue 118 to 240 (cyan, Uniprot: Q07444). The zoomed-in pictures depict the structural discrepancies in the loop (top) and the following β-sheet (bottom) between KLRC2 and KLRC3. **c** Predicted structures of KLRC/HLA-E/CD94 complex (KLRC2 from residue 118 to 231: orange, Uniprot: P26717; KLRC3 from residue 118 to 240: cyan, Uniprot: Q07444; Full-length HLA-E: purple, Uniprot: I3RW89; CD94 from residue 57 to 179: green, Uniprot: Q13241). The zoomed-in protein structure depicts the interaction interfaces of KLRC2/HLA-E (top) and KLRC3/HLA-E (bottom)

be under positive selection with greater 90% possibility by the branch-site model implemented in PAML ($p = 0.006$, likelihood ratio test) (Fig. 5b, Additional file 2: Table S13).

Based on the AlphaFold2 and KLRC1 crystal structure [53, 65], the protein structure of KLRC2 and KLRC3 are predicted to be altered by these three amino acids (Fig. 5b). KLRC proteins have been shown to bind CD94 and HLA-E for immune response [53, 65]. With predicted complex protein structure of KLRC/HLA-E/CD94, significant differences were found between the interaction interfaces of KLRC2/HLA-E and KLRC3/HLA-E. Two hydrogen bonds were observed between the C-terminal of KLRC3 and HLA-E: the amide nitrogen of Ile226 from KLRC3 binds the side chain of Glu175 from HLA-E; the side chain Arg227 from KLRC3 binds the carbonyl oxygen of Asp170 from HLA-E. The two hydrogen bonds may stabilize the flexible loop of KLRC3 (Fig. 5c). However, no obvious interactions were found between the C-terminal of KLRC2 and HLA-E (Fig. 5c). Our findings implicate differential binding affinity at the two interaction interfaces potentially important for functional differentiation of KLRC2 and KLRC3 in humans.

## Discussion

The first complete genome assembly (T2T-CHM13) is an important genomics resource [4–9, 11, 66, 67]. Here, we investigate the large-scale genomic differences between T2T-CHM13 and the current reference genome assembly (GRCh38). We show that the discrepant regions are among the most structurally complex and may introduce reference biases in human genetics (e.g., genotype–phenotype association study) and evolutionary genomics (e.g., gene family evolution investigation). Therefore, understanding the discrepant regions between the two crucial reference genome assemblies will provide a crucial resource for further genomic and functional studies. In this study, we systematically characterized the large SVs between the two human genome assemblies and found 67 newly identified discrepant regions. In addition, we developed an integrated website tool (SynPlotter) to visualize and validate 238 discrepant regions. The newly identified regions include gene-model differences (e.g., *ZDHHC11B*, *GSTM1*, *CFHR3*, *CFHR1*, *CR1*, and *KLRC2*) and the SGDP read-depth genotyping data show that the discrepant regions are more likely CN polymorphic. In addition, the discrepant regions are often related to human diseases. Finally, we provided a novel example to illustrate the biological importance of discrepant regions by analyzing the *KLRC* gene cluster with population genetics and evolutionary genomic approaches.

Previous studies used a 1Mbp syntenic intervals to identify the sequence difference between the two genome assemblies [4, 9]. Here, we used three different methods to identify the SVs ($\geq 10$ kbp) with reciprocal alignment between GRCh38 and T2T-CHM13 to identify precise breakpoints and structural types of the large-scale discrepant regions (Additional file 1: Fig. S1). We additionally identified the 67 discrepant regions and ~70% of them belongs to deletions, because deletion variants are likely chained in a large synteny by the LASTZ [4, 9]. Notably, comparing with the recent inversion dataset in humans [68], we found that one inversion in our dataset was not reported in the dataset [68]. The genomic region of the inversion contains a gap in GRCh38 (Additional file 1: Fig. S17). This shows the SV discovery would be affected by the reference bias and T2T-CHM13 is useful to identify large-scale SVs. In addition, we developed an integrated visualization tool to validate the discrepant regions. This website tool is user-friendly and publicly available to compare syntenic regions between GRCh38 and T2T-CHM13.

The discrepant regions between these two assemblies have been regarded as CN polymorphic genomic regions in previous studies [4, 6, 9]. We performed CN analysis to provide clear evidence to support that the discrepant regions are likely more polymorphic than the genome-wide average, even than the CN variable regions. In addition, to our knowledge, we surveyed the relevance between the discrepant regions and the reported medical relevant loci in greater detail. We find that rare microdeletions and microduplications of 27 discrepant regions are potentially related to neurodevelopmental diseases and others with supported evidence [29–31]. Loss of function of *CR1* is associated with Alzheimer's disease [36, 37] and T2T-CHM13 carries a major allele of *CR1*. Yet, GRCh38 carries a minor allele, where eight exons encoding tandem repeat protein domain in *CR1* are inserted with respect to T2T-CHM13. In addition, *ZDHHC11* (Zinc Finger DHHC-Type Containing 11) and *ZDHHC11B* are involved in innate immune or anti-virus response by enabling signaling adaptor activity. The CNV of *ZDHHC11* and *ZDHHC11B* are associated

with hepatoblastoma [69] and primary open-angle glaucoma [70]. The *GSTM1* (glutathione S-transferase mu) locus is also a polymorphic locus associated with cancers, metabolism, and hepatic cirrhosis [71]. The *GSTM* recurrently mutated during primate evolution with syntenic comparison and phylogenetic analyses (Additional file 1: Fig. S18-19, Additional file 2: Table S14). Thus, our study provides a fundamental resource for functional assessments to examine functional differentiation between/among polymorphic loci in humans. Importantly, it is still unclear whether the reference bias has effects on the reported disease association study of these discrepant regions. If so, the excess of the co-localization between the discrepant regions and disease-relevant CNV needs to be re-assessed. In addition, other studies on duplicated genes show that non-essential, less important, or fast-evolving genes are more likely to be duplicated during evolution [72–75]. Therefore, functional assessments of these polymorphic loci are worth considering in future studies.

We comprehensively compare the *KLRC* gene cluster in humans and NHPs. Firstly, we precisely characterize the breakpoints of duplication and deletion of *KLRC2.* The breakpoints on single-base-pair resolution could facilitate the molecular probe development to genotype the CN of *KLRC2* in the future. The duplication and deletion mechanism of *KLRC2* are associated with the *Alu* elements and simple repeats in the human genome. Notably, our *KLRC* haplotype inference and phylogenetic tree analyses show that the origin of *KLRC2* and *KLRC3* is duplicated from the common ancestor of the apes and Old-World monkey. The human population genetic analyses reveal that *KLRC*-hap2 (*KLRC2* depletion) is caused by a single-deletion event in humans. Africans and Asians have a higher frequency of *KLRC*-hap2, but we did not observe significant pi diversity change in the *KLRC* gene cluster in humans. These results would suggest that the distribution of the *KLRC* haplotypes may simply be the result of genetic drift in human evolution. Yet, we identified the six SNVs to distinguish *KLRC*-hap2 (*KLRC2* depletion) from *KLRC*-hap1. The eQTL and GWAS analyses show the gene expression and immune functional differentiation between the two *KLRC* haplotypes, and previous functional experiments show the *KLRC2* haplotypes have different roles in synaptic pruning [54]. Additional experiments are required to determine if loss of *KLRC2* is the result of genetic drift or subject to other models of selection (e.g., balancing selection).

Our tests of selection implicate three amino acids of *KLRC3* as potentially subject to positive selection during primate evolution. Predicted protein structures further suggest structural differences (KLRCX/HLA-E/CD94) between KLRC2 (KLRC2/HLA-E/CD94) and KLRC3 (KLRC3/HLA-E/CD94). It is possible that *KLRC3* has acquired distinct functional properties from *KLRC2* as a result of natural selection. We also show that the KLRC gene cluster region was misassembled in GRCh38, since the region was assembled by two BAC clones from two distinct KLRC haplotypes. If we used the six SNVs to infer the KLCR haplotype, GRCh38 would carry KLRC-hap2 (KLRC2 deletion). Yet, the GRCh38 shows KLRC-hap1 at present. As a result, association studies of KLCR genes and their interpretation would be potentially confounded.

## Conclusions

Altogether, our study provides a more comprehensive and detailed assessment of the structure and function of the large-scale discrepant genomic regions between GRCh38 and T2T-CHM13. We believe the results of this work not only contribute to

our biological understanding of these diverse regions but will benefit future studies by helping to eliminate reference biases. We should stress that our study focused solely on the large-scale discrepant regions between two "completed" genome assemblies and, as such, represents limited sample sizes [18, 19]. We expect that the HPRC, CPC, and T2T Consortium will generate more complete genome assemblies from a diversity of samples [26, 76]. These will help us to fully understand the extent of complex/discrepant regions in humans [4, 11, 25, 26] and their biological impact using reference-free approaches. In addition, we explored the *KLRC* gene family in detail, but it was not possible to examine the evolutionary history of each discrepant region in this study. Furthermore, we anticipate that we will gain a better understanding of the evolutionary history of each base in the human genome with the availability of complete primate genomes in the near future.

## Methods

### Data in this study

We downloaded 94 long-read human genome assemblies from the HPRC phase 1 project (https://humanpangenome.org/) [24–26]. We download the Illumina data of the 2504 high-coverage short read from the 1 KG human population dataset [59]. For the reconstruction of the phylogeny of the *KLRC* gene cluster, we locally assembled the *KLRC* gene cluster region from the published HiFi reads of chimpanzee, bonobo, gorilla, orangutan, gibbon, macaque, owl monkey, and marmoset. In addition, the "non-syntenic" region, centromere, and gene annotation files were downloaded from the UCSC Genome Browser directly (https://genome.ucsc.edu). The (sub)telomere regions are defined as a 500-kbp region away from the start or end of chromosome in this study.

### Discrepant region characterization and validation

We used a reciprocal alignment approach to systematically characterize the SVs. In detail, we used GRCh38 as the reference genome and T2T-CHM13 as the query to run PAV (v2.0.0) [18, 19], PBSV (PBSV, v2.8.0, https://github.com/PacificBiosciences/pbsv), and minigraph [20] (https://github.com/lh3/minigraph, commit 86192499e80377df-47993cb376e4773d4a7a76db) to characterize SVs. We also used T2T-CHM13 as the reference and GRCh38 as the query to run PAV, PBSV, and minigraph to characterize SVs. Then, we LiftOver the coordinates from GRCh38 to T2T-CHM13 and then merged these calls with bedtools (v.0.29.0). In the PBSV analysis, we used the PBSIM2 tool [77] to simulate HiFi reads from GRCh38 and used these simulated reads against T2T-CHM13.

We developed a custom script to automatically screenshot the syntenic plots from SafFire (https://mrvollger.github.io/SafFire). In addition, we integrated minimap2 (v2.24) [21] or mummer4 aligner [22] to generate syntenic PAF files. We next applied the dot-plot implement in mummer4 [22] to generate dot plots. Then, we implement the above scripts into a website tool (SynPlotter) to visualize the syntenic relationship between two coordinates. With our website tool, the syntenic plot and dot plot can be generated, and the basic genomic difference statistics could be calculated. The gene and repeat (e.g., segdups) annotations are also shown. Lastly, we used our website tool to validate the large SVs ($\geq 10$ kbp) generated by the three above callers. In this study, we defined structural types by detecting SVs on T2T-CHM13 with respect to GRCh38. Deletion refers

a genomic region that is absent in T2T-CHM13 with respect to GRCh38, whereas SDR refers to a complex genomic difference rather than a simple deletion/insertion/inversion.

### Gene, structural type, and repetitive sequence annotation for discrepant regions

We used bedtools (v2.29.0) to compare the discrepant regions between "non-syntenic" regions and our large SVs ($\geq 10$ kbp). We used the ggplot2 [78] and karyoploteR packages [79] to plot the chromosome ideogram. Next, we characterized the discrepant regions into four types (insertions, deletions, inversions, and SDRs) with eyes and manually refined the breakpoints of these SVs.

We also used the gene, repetitive sequence annotation files from the UCSC Genome Browser to annotate these discrepant regions with bedtools (v2.29.0) [80]. In this study, the centromere regions include pericentric regions, but not the centromeric transition regions in all analysis. Notably, the previous study reported sequence difference between GRCh38 and CHM13, but our study reported the location of the discrepant regions. In Fig. 1, we counted the number and the length of the discrepant regions as centromere (CEN), segdup (SD), telomere (TEL), and tandem repeat (TRF) based on the location of them, not the absolute length of sequences belonging to each type.

We used a hierarchy approach to define SV-location: (1) If a SV located in CEN, we counted it as CEN; (2) If a SV located in TEL (500kbp from head or tail), we counted it as TEL; (3) If a SV located in segdup and includes at least 20% or 2kbp segdup sequence, we counted it as SD; (4) If a SV located in TRF and includes at least 10% or 1kbp sequence, we counted it as TRF; (5) If a SV does not belong to any type of above, we counted it as others. For example, if a SV located in the centromere regions, we counted it as CEN type no matter whether it contains segdup sequence or not.

### Structural polymorphism enrichment analysis

To test whether the discrepant regions are more likely polymorphic, we downloaded the SGDP CN table [17] from the UCSC Genome Browser (https://genome.ucsc.edu/). Here, to reduce the bias from the high CN (average CN estimation from SGDP $\geq 10$, $n = 53$), we only used the 131 discrepant regions (CN < 10) belonging to insertions, deletions, and SDRs to calculate the standard deviations (s.d.) of the CN. First, we used bedtools (v.2.29.0) to intersect the 131 discrepant regions with the SGDP CN table and then calculated the s.d. of the CN of each intersected region. Then, we calculated the mean and median values of estimated s.d. (mean = 0.735, median = 0.439). These are our observed s.d. values of the 131 discrepant regions.

For null distribution, we did two experiments in this study. (1) Simulate the distribution of mean s.d. of the whole genome. We used bedtools (v2.29.0) to randomly shuffle the corresponding number of coordinates ($n = 131$) in the genomic regions where there are no centromeres, telomeres, and CN < 10. We intersected them with the SGDP CN table, and calculated the s.d. of the CN for each intersected region. Then, we calculated the mean/median s.d. value. (2) Simulate the distribution of mean s.d. of the CNV regions (CN > 2.5 and CN < 10). We used bedtools (v2.29.0) to randomly shuffle the corresponding number of coordinates ($n = 126$) in the genomic regions where there are no centromeres, telomeres, CN > 2.5, and CN < 10. We

intersected them with the SGDP CN table and calculated the s.d. of the CN for each intersected region. Then, we calculated the mean/median s.d. value.

We repeated this 1000 times for each experiment and calculated the empirical *p*-value of our observed mean s.d. value (permutation test). In addition, we also estimated the observed mean and median s.d. values of the SGDP CN table for the two experiments. (1) We calculated the mean and median s.d. value of the regions (CN < 10) based the SGDP CN table (mean = 0.13, median = 0.079). (2) We calculated the mean and median s.d. value of the regions (2.5 < CN < 10) based on the SGDP CN table (mean = 0.58, median = 0.463).

### Disease-relevant CNV enrichment and survey

We downloaded the CNV coordinates from the morbid and the cross-disorder dosage sensitivity maps [29–31] and LiftOver them to T2T-CHM13. We next used bedtools to intersect our discrepant regions with the integrated coordinates and found 33 discrepant regions are co-localized with the disease-relevant CNVs. We also used a permutation test to shuffle the discrepant regions in T2T-CHM13, excluding centromeres and telomeres, and calculated how many discrepant regions could be co-localized with the disease-relevant CNVs. We repeated this process 1000 times and plotted the distribution of the number of co-localized regions (mean $N = 19.9$) with ggplot2 in R.

We also manually curated the coordinates with gene annotations from the literatures and DECIPHER database by hands. To better represent the disease-relevant discrepant regions, we only listed the regions with well-qualified evidence/literature/case-report to support as disease relevant in Table 1.

### Genomic syntenic comparison analysis

In this study, we found 67 newly identified discrepant regions, of these, the 25 regions contain 38 genes. Thus, we used minimiro (commit 18271297374ae6a679521a7ce3f5bb6c0cf8d261) to compare the genomic syntenic relationships between GRCh38 and T2T-CHM13 in these regions.

Then, we used the RefSeq annotation from GRCh38 and CAT/RefSeq annotation from T2T-CHM13 to extract the protein sequences of the genes. The mafft program (v7.4.3) [81] was used to align the amino acid to check the amino acid difference among the homologous genes. The schematic plots were generated by ggplot2 and AliView (v1.26) [82].

### KLRC2 haplotype characterization

We extracted the genomic regions containing the *KLRC* gene cluster region from GRCh38 (chr12:10359648–10,470,462). Then, we used minimap2 (v2.24) to map the region to 94 long-read human genome assemblies and other NHP long-read genome assemblies. Finally, we used the minimiro to generate the syntenic plots between GRCh38 and other human and NHP samples. We found three distinguished haplotypes of the *KLCR* gene cluster based on the CN variation of *KLRC2.*

### *KLRC* haplotype inference from 1 KG population dataset

We used the previously reported read-mapping approach (SUNK-WSSD) [55, 56] to genotype the CN of *KLRC2* and *KLRC3* in the 1 KG population dataset (2504 high-coverage Illumina genomes). The mean CN of *KLRC3* of 2504 humans is ~ 1.8 (s.d.: 0.14) (Additional file 1: Fig. S6), while the mean CN of *KLRC2* of 2504 humans is ~ 1.9 (s.d.: 0.6). The CN of *KLRC3* is clustered together in different human groups, suggesting there is no CN variation of *KLRC3* in humans (Additional file 1: Fig. S20). However, the CN of *KLRC2* is clustered into three groups in different human groups, suggesting there is CN variation of *KLRC2* in humans.

If the CN of *KLRC2* is greater than 2.5 (mean + 1 s.d.), we inferred the *KLRC2* CN as 3. If the CN of *KLRC2* is less than 1.3 (mean − 1 s.d.), we inferred the *KLRC2* CN as 1. Then, we used the maps, ggplot2, and scatterpie packages (https://www.rdocumenta tion.org/packages/scatterpie/versions/0.1.8) in the R to plot the world map of the *KLRC* haplotype map.

### Phylogeny reconstruction and time-calibration tree reconstruction

We used minimap2 (v2.4) to determine the syntenic regions in human and NHP genome assemblies. We also used samtools (v1.9) to extract the corresponding regions. Then, we used mafft (v7.453) to align the genomic sequence with default parameters and used it as input for IQTREE (v1.6.11) to build the maximum likelihood phylogenetic trees [21, 83–85].

Here, we used the log-normal and the real mean model to set the prior calibrate time, including pan-lineage split time (~ 1.45 mya), owl monkey and marmoset split time (~ 24.5 mya), and monkey and ape split time (~ 54 mya) in this study. The divergence time was estimated using the HKY substitution model, relaxed log-normal clock model, and calibrated Yule prior with the divergence time described above. The MCMC chains were run 30,000,000 steps, and 3,000,000 steps were set for burin running. Finally, we used the tracer (v1.7.1) to examine whether the chain was convergent. Indeed, each ESS value of each parameter was over 200 in our study and these results suggested the MCMC chain was converged. We repeated this divergence time estimation three times independently, with each run converging and producing similar estimated times. All results are available on our GitHub page (https://github.com/YafeiMaoLab/discrepant_ region.git).

### Selection test with PAML and aBSREL

We downloaded the human and NHP coding sequences (CDS) and protein sequences for *KLRC2* and *KLRC3* from the UCSC Genome Browser. We used mafft (v7.4.3) to align the protein sequences and used translatorx_vLocal.pl to align the CDS based on the aligned protein sequences. All protocols are based on the previously reported tool (TREEasy).

We examined the pi diversity of the *KLRC* gene cluster regions in humans with 94 long-read genome genotyping data. Then, we ran a preliminary selection test on aBSREL [64] (https://www.datamonkey.org/analyses) and the aBSREL tool showed the selection signals on the *KLRC3* clade. Specifically, we used the CDS alignment as input and selected the *KLRC3* clade branch for testing selection pressure with a full adaptive

model. After a *p*-value correction, the aBSREL analysis with the full adaptive model revealed significant selection pressure in the *KLRC3* clade branch ($p=0$, Additional file 1: Fig. S16). We also ran the branch model in PAML (v4.9) [63, 86] and the model showed the selection signals on the *KLRC3* clade too ($p=0.03$). The branch-site model in PAML (v4.9) shows three amino acids under selection with a probability greater than 0.9 in the clade of KLRC3 in the Bayes Empirical Bayes (BEB) analysis ($p=0.006$). The *p*-values were calculated by the likelihood ratio test in R.

In the branch model, we set the following parameters to establish the null model: 'runmode$=0$, seqtype$=1$, CodonFreq$=2$ (F3X4), model$=2$, NSsites$=0$, getSE$=0$, icode$=0$, fix_kappa$=0$, kappa$=1$, fix_omega$=1$ (omega fixed), and omega$=1$.' We assumed that all branches have an omega value of 1, with $np=13$ degrees of freedom. To test whether the *KLRC2* and *KLRC3* clades have different selection pressure (omega values), we utilized the same parameters as the null model, but with different user-specified trees and free dN/dS ratio set for the two clades, with $np=13$ as described in a previous study [87]. We observed that the *KLRC3* clade with the estimated omega model has a lower likelihood value ($p=0.033$), indicating that the clade is not evolving neutrally.

In the branch-site model, we used the following parameters to set the null model: runmode$=0$, seqtype$=1$, CodonFreq$=2$ (F3X4), model$=2$, NSsites$=0$, getSE$=0$, icode$=0$, fix_kappa$=0$, kappa$=1$, fix_omega$=0$ (omega free), and omega$=1$ (initial omega). This assumed that all branches have a free omega with $np=14$ (degrees of freedom). To test for different selection pressures on different sites in the *KLRC3* clade, we used the following parameters: runmode$=0$, seqtype$=1$, CodonFreq$=2$ (F3X4), model$=2$ (user-specified dN/dS ratios for branches), NSsites$=2$, getSE$=0$, icode$=0$, fix_kappa$=0$, kappa$=1$, fix_omega$=0$ (omega free), and omega$=1$ (initial omega), with $np=16$ as described in a previous study. We observed that three amino acids have a possibility of being under positive selection greater than 0.9 in the *KLRC3* clade ($p=0.006$), suggesting that these amino acids are likely under positive selection.

Our selection tests may be affected by several factors, including GC content, saturation, and small sample size [88–90]. To address these potential issues, we examined the GC content (GC 40–42) and the saturation level (tree length 0.52) in our empirical data, and the data fit for the model (Additional file 2: Table S15). We used Fisher's test with a small sample model to test for selection pressure (Additional file 2: Table S16), and the results were condicent with those obtained using PMAL.

### Protein structure analysis of KLRC2 and KLRC3

We predicted the structural model of KLRC2 (residues 118–231) and KLRC3 (residues 118–240) using AlphaFold2 [91] and KLRC1 crystal structure [53, 65] with predicted local distance difference test (pLDDT) values as 92.74 and 80.55, respectively, suggesting that they are accurate enough for the further analysis. AlphaFold2 [91] was used to predict the structures of NKG2-C/MHC/CD94 and NKG2-E/MHC/CD94 complexes. The pLDDT values of the two complexes are 80.39 and 76.05, respectively, which are of high confidences and are accurate enough for the interaction analysis. Protein structure and interaction analyses were performed on PyMol (v2.4.1, https://pymol.org/). Structure alignment shows obvious differences between KLRC2 and KLRC3. Met223 of KLRC2 and Arg224, Arg227, and Gly229 of KLRC3 are located at the surface loops

which connect two β-strands, and the loop of KLRC3 has a longer conformation. His226 and Lys228 of KLRC2 may contribute to the following β-strand, which is longer than that of KLRC3.

### eQTL analysis and GWAS ATLAS analysis

We firstly aligned the *KLRC* gene cluster of 94 long-read human genome assemblies and we used our custom script to find the SNVs that are different between *KLRC*-hap2 and *KLRC*-hap1. Then, we investigated LD among Lewontin's D′ and R2 implemented in LDBlockShow (v1.40) [92] and PLINK (v1.09) [93] with 2504 high-coverage genotyping data from the 1 KG dataset. The SNVs with minor allele frequencies > 10% were used for this analysis. We also calculated the allele frequency of the six distinguished SNVs in the gnomAD dataset (v.3.1.2, https://gnomad.broadinstitute.org/) [94]. The LD heatmaps were generated by LDBlockShow or R.

We also used PLINK (v1.90b6.21) [93] to compute LD (measured with D′) among all SNV pairs. For each SNV pair, we then compared the reference alleles to the combinations of alleles that were determined to be "in phase" (i.e., observed together on haplotypes more often than expected under linkage equilibrium). For cases where the reference genome carried alleles that were in phase, D′ was retained as a positive value, whereas for cases where the reference genome carried alleles that were out of phase, D′ was multiplied by − 1 to indicate that the alleles are in repulsion. The same procedure was repeated for the corresponding region of T2T-CHM13 as determined by LiftOver, using genotype data produced by Aganezov et al. [6]. For comparison, we also performed the same analysis for a randomly selected "control" region of the same length (83.7 kbp) for both GRCh38 and T2T-CHM13.

We downloaded the eQTL multi-tissue data from GTEx (release v8, https://gtexportal. org/) and we extracted the gene expression associated with the six SNVs in different tissues. The data showed that the six SNVs are only associated with *KLRC2* gene expression, as we expected. Then, we used the ggplot2 package in R to plot the normalized effect size and *p*-values of gene expression difference by the six SNVs. In addition, we also investigated whether any locus is related to the reported genome-wide association study (GWAS). Then, we download the phenome-wide association studies (PheWAS) data from GWAS ATLAS [60]. The data showed that three of the six SNVs are significantly associated with NK cells (NKearly: %335 + 314- and NKeff: %314- R7-) [45].

### Supplementary Information

**Additional file 1: Fig. S1.** The visualization of four discrepant regions bydotplot. **Fig. S2.** Comparison of the gene length of *CR1* in long-read human genome assemblies. **Fig. S3.** Gene structure differences in the discrepant regions. **Fig. S4.** The discrepant region contains *CFHR1* and *CFHR3*. **Fig. S5.** The number of gene difference in deletions (total $N = 68$) and insertions (total $N = 87$) between GRCh38 and T2T-CHM13. **Fig. S6.** The distribution of the median CN s.d. **Fig. S7.** The distribution of intersected disease-related CNVs. **Fig. S8.** The SGPD read-depth genotyping of *KLRC2* in different human populations. **Fig. S9.** The haplotype structure of living human populations in T2T-CHM13. **Fig. S10.** SNV distribution in T2T-CHM13 and GRCh38 of two chromosome regions. **Fig.S11.** CN of *KLRC2* and *KLRC3* for NHP population level. **Fig. S12.** The syntenic relationship of *KLRC* gene cluster region between human and different apes. **Fig. S13.** The syntenic relationship of *KLRC* gene cluster region between human and monkey genomes. **Fig. S14.** The phylogenetic tree of *KLRC2* and *KLRC3* with mouse lemur as outgroup. **Fig. S15.** The pi diversity of the *KLRC* gene cluster based on the 94 long-read genome assemblies. **Fig. S16.** Selection pressure testing using aBSREL on *KLRC3* clade. **Fig. S17.**Comparison of the genomic region of the inversion containing a gap in GRCh38. **Fig. S18.**

The syntenic relationship of *GSTM* gene cluster region between human and NHPs. **Fig. S19.** The phylogenetic tree of *GSTM* in primates. **Fig. S20.** The CN of *KLRC3* in different human populations.

Add**itional file 2: Table S1.** The large-scale SVs discovered by whole-genome alignment (PAV). **Table S2.** The large-scale SVs discovered by graph-based (minigraph). **Table S3.** The large-scale SVs discovered by read-mapping (PBSV). **Table S4.** The 238 validated large SVs. **Table S5.** The summary table of the disease-related or gene-related discrepant regions. **Table S6.** The gene differences between the two human reference genomes in 155 validated insertion and deletion large SVs. **Table S7.** Genotyping insertions and deletions on HPRC and HGSVC assemblies. **Table S8.** The GO enrichment of the genes in the 27 disease-related discrepant regions. **Table S9.** The three *KLRC* haplotypes identified from the 94 long-read genome assemblies. **Table S10.** The frequency of *KLRC* haplotype in the 94 long-read assemblies and in the inferred 1KG dataset. **Table S11.**The frequency of *KLRC*-haps in the different populations. **Table S12.**The frequency of the six SNVs in GnomAD and 1KG database. **Table S13.** The likelihood of the branch/branch-site models in PAML. **Table S14.** Copy number difference of protein-coding and SV-affected genes between human and non-human primates. **Table S15.** GC content of *KLRC2&3* in different species. **Table S16.** The selection test on small sample size model.

**Additional file 3.** Review history.

### Acknowledgements
We thank T. Brown for assistance in editing this manuscript. We thank P. Hsieh and H. Cheng for their comments and scripts. We thank the HPRC for providing the 94 long-read human genome assemblies. The computations in this paper were run on the Siyuan-1 and π 2.0 cluster supported by the Center for High Performance Computing at Shanghai Jiao Tong University.

### Review history
The review history is available as Additional file 3.

### Peer review information

### Authors' contributions
Y.M. conceived the project. X.W., X.Y., and Y.M. finalized the manuscript. X.W. and Y.Z. performed the SVs analysis. X.W., Y.Z., and X.Y. performed the analysis of large discrepant regions. X.W., X.Y., S.Z., L.F., and Y.M. performed the *KLRC* gene cluster analysis. M.X., Q.L., and Y.M. performed the KLRC protein structure. Y.M., D.J.T., R.C.M., and M.C.S. performed the *KLRC2* haplotype swap analysis in GRCh38. S.Z. built the "SynPlotter" website. Y.M., X.Y., X.W., M.X., L.F., M.R.V., N.C., W.T.H., G.A.L., D.M., J.S., R.C.M., M.C.S., W.L., Q.L., and E.E.E. contributed to interpret results and edited the draft manuscript. All authors read, edited, and approved the manuscript.

### Funding
This work was supported by Shanghai Pujiang Program (22PJ1407300) and Shanghai Jiao Tong University 2030 Program (WH510363001-7) to Y.M.; and by National Natural Science Foundation of China (82001372) to X.Y. This work was supported by Opening research fund from Shanghai Key Laboratory of Stomatology, Shanghai Ninth People's Hospital, College of Stomatology, Shanghai Jiao Tong University School of Medicine (Grant No. 2022SKLS-KFKT007) to Y.M. This work was supported, in part, by US National Institutes of Health (NIH) grant# HG002385 to E.E.E. Additional funding included NIGMS grants: K99GM147352 (to G.A.L.) and National Natural Science Foundation of China grant: 32000812 (to J.S.). E.E.E. is an investigator of the Howard Hughes Medical Institute.

### Availability of data and materials
The website tool SynPlotter can be accessed at https://synplotter.sjtu.edu.cn/. Mummer4 and SafFire (https://github.com/mrvollger/SafFire) are dependencies for SynPlotter [9, 22, 95, 96]. The HPRC data used in this study can be found in the public database [24–26]. The syntenic comparison of discrepant regions is deposited in GitHub (https://github.com/YafeiMaoLab/discrepant_region.git) [95]. The code utilized for data analysis and benchmarking of background methods is also available on GitHub at https://github.com/YafeiMaoLab/discrepant_region [95] under the MIT license. Additionally, it has been deposited in Zenodo under the https://doi.org/10.5281/zenodo.8058461 [96].

## Declarations

### Ethics approval and consent to participate
Not applicable.

### Consent for publication
All authors have read and approved the submission of this manuscript.

### Competing interests
E.E.E. is a scientific advisory board (SAB) member of Variant Bio. N.C. is a full-time employee of Exai Bio. The other authors declare that they have no competing interests.

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

## 