## [**Additional file 3. **Review history. · Genome Biology]

Review History

First round of review

Reviewer 1

Were you able to assess all statistics in the manuscript, including the appropriateness of statistical tests used? No, I do not feel adequately qualified to assess the statistics.

Comments to author:

Yang et al., report on the large-scale discrepant region in the T2T compared to hg38. While this analysis has been performed previously, the authors identify some previously unidentified non-syntenic regions which is of utility to the genomics community. They deploy a novel web-based tool (Synplotter) to characterise these regions and identify KLRC as a cluster with potential functional relevance. While I do not have experience with the bioinformatic tools deployed here the analysis appears to be robust and well supported by orthogonal short-read and long-read data. The paper is clearly written and presented and unusually I did not find any faults with the paper. The novel insights are perhaps somewhat modest, but given the importance of accurate reference genomes and identifying structural differences between individual genomes I believe it merits publication.

Reviewer 2

Were you able to assess all statistics in the manuscript, including the appropriateness of statistical tests used? Yes, and I have assessed the statistics in my report.

Comments to author:

Yang et al. present a study comparing two important human genome assemblies: the current human genome reference (GRCh38) and the first telomere-to-telomere (T2T) human genome assembly (T2T-CHM13). The study identifies 590 discrepant regions between the two assemblies, which are highly structurally polymorphic in humans and likely associated with various human diseases. The study specifically focuses on a newly identified discrepant region, the KLRC gene cluster, which is associated with natural killer cell differentiation. And one gene KLRC3 has undergone rapid amino acid replacements during primate evolution. This study helps to further our understanding of the large-scale structural variation differences between these two crucial human reference genomes and their implications for the interpretation of studies of human genetic variation.

The paper is clearly written and the results are solid given the application of variety of datasets as sort of cross validation. I only have several suggestions:

1. The genes identified in the discrepant regions seem from deletion events with respect to GRCh38, correct? Since only several genes were identified, it would be great to try to identify additional genes, if any, in other discrepant regions such as the insertion regions. A gene prediction tool could be applied or compare to some outgroups.
2. Because the discrepant regions are usually involved in complex genomic architectures (SD, CEN, TEL, TRF, etc.), it is challenging to genotype these regions. The application of read-depth genotyping

from Illumina short reads data is potentially questionable - as reflected by the copy numbers range from 0 to over 10. The authors mentioned 94 HPRC long reads dataset. HGSVC also released 64 haplotype resolved genomes (Ebert et al. 2021). It would be great to genotype these regions comprehensively using these long reads. Based on the long-read results or pangenome results, PanGenie may be applied to accurately genotype these regions.

3. The authors identified approximately 226 Mbp discrepant regions. However, only the KLRC gene cluster looks significant and half of the manuscript is discussing this cluster, though very interesting. The molecular evolution analysis is very interesting. If the analysis can be applied to all the discrepant regions or at least the euchromatic regions, then it would definitely promote the work further. For example, the authors could simply check the divergent time along the non-human primate lineage for these regions. Are they human specific or common between species?

4. The GitHub page contains the visualization results of the 238 discrepant regions. It would be great to include all the scripts and codes of the results generated in the manuscript. This will largely increase the reproducibility of the work.

Reviewer 3

Were you able to assess all statistics in the manuscript, including the appropriateness of statistical tests used? Yes, and I have assessed the statistics in my report.

Comments to author:

Yang and colleagues use the recent T2T human genome build to look for euchromatic regions that are new or discrepant from the standard GRCh38 build identifying 238 such regions, 67 of which were not previously recognised. They provide a useful tool to aid this. They also go beyond simple annotation as "non-syntenic" as done originally to characterize different modes of difference (insertions, deletions etc). They go on to examine gene models in the discrepant regions finding high copy number polymorphism rates. They also identify amino acid level differences in these genes suggesting polymorphism for both copy number and sequence. This motivates a call for investigation of these as possible polymorphisms associated with disease or unusual evolutionary patterns. They then move focus to consider a newly identified discrepant region gene KLRC identifying three haplotypes in humans (with 0,1, and 2 gene copies of KLRC2). Interestingly, the T2T genome has a KLRC2 deletion. They identify duplication of KLRC2 and KLRC3 at the LCA of apes and old-world monkeys. They identify 3 amino acids in KLRC3 that appear to have undergone positive selection. They identify a deletion in KLRC2 in 20% of humans. This appears to be immune related (a known functionality associated with rapid evolution). From this they infer biological importance of the discrepant regions.

This is a very impressive and sizeable body of research (and the above doesn't capture all the results!). Indeed, so sizeable that one must question whether this is in fact one paper. At one level it is very broad scale. At the end we consider three amino acids in one protein and details of protein structures and interactants. As is there doesn't seem to be an easy to grasp story here. This is rather reflected in the last line of the abstract that claims rather vaguely

that the work "furthers our understanding...". Well, true it does, but it is revealing that nothing more specific can be claimed.

Indeed, I would say I found this an especially difficult paper to review because of the changes in scale and absence of focus. Fortunately, I was asked to specifically consider the evolutionary aspects and so will restrict my comments to these issues. This being said, I was routinely confused as to whether the differences between the complete and incomplete builds are owing to incompleteness of the original build or bona fide variation between genomes in certain highly variable regions. If the latter then you don't need T2T you just need long reads and well built genomes. From what I can see the variation in the euchromatic sections seem to largely be the latter (suggested on line 158, CN polymorphism seems to be associated with discrepant regions) not the former. The interesting analysis of the misbuild of the KLRC region in GRCh38 was however also rather instructive.

If the claim is that variable regions are interesting because they are variable then this needs much better framing. In the intro the authors argue that dynamic regions are important regions for both adaptation and disease (page 3 line 69). This strongly reflects old claims that after gene duplication duplicates evolve faster owing release from constraint and neo-functionalisation. However, more recently a more interesting possibility has been shown to be more likely: genes that change copy number and are fast evolving, are like this because they are unimportant. They always were fast evolving, their knockouts tend to have no phenotypes and they are possibly just neutral tolerated duplications 1-4. It would be helpful to frame the CNV and gene duplication in a manner that allows for these more recent possibilities. The idea that these variable regions may be neutral is touched on in the discussion, but not framed wrt this literature and evidence base either there or in the introduction.

For the sections I would like to comment on, and have looked at in some detail, I decided that I cannot really comment on the results as the methods are so poorly described that I don't know what has been done and assumed. Normal reporting standards have not been followed. Stress testing has not been performed. I think the general approach may be sound (I can't really say) but certainly not replicable. This is a common problem with expansive multi-story papers. I highlight:

* At the individual gene level evolution we are told that PAML was used with a branch and a branch-site model (Table S11). This is fine but in a paper on positive selection one would usually give many more details: what parameters were specified in each (codon model?), what degrees of freedom do you assume when comparing likelihoods of the nested models? What are the nested models? As is we are just presented with tree topologies, likelihoods and P values. Near uninterpretable I would suggest.

* The aBSREL analysis (line 300). As the aBSREL authors note there is an optimum complexity for BS models. What did you use and where are the results? Table S11 says it is just PAML. I'm confused.

* Fig 5a uses BEAST2 - input parameters, how run?

* Are the PAML results robust to modifications? I note that the Branch models aren't overwhelmingly significant, but I also have no idea what you did. Branch-site models in particular are also rather notoriously prone to false positives, problems with saturation and GC content^{5,6} so stress testing the results is important.

* Fig S13 you claim no significant differences based on pi - line 298, method line 572. Differences between what and what? What test did you do? Second, you should control for local recombination rate (diversity is expected to be higher when recombination is lower owing to background selection).

These are just some of the issues. In sum, this looks like a data/analysis rich paper with lots of potentially interesting results but I'm not sure it is one paper. Multiple much better described papers may be preferable (the KLRC story could be a standalone). As is, this is not replicable with results and methods more akin to late period Picasso than an exquisitely detailed Vermeer.

1 O'Toole, A. N., Hurst, L. D. & McLysaght, A. Faster Evolving Primate Genes Are More Likely to Duplicate. *Molecular Biology and Evolution* 35, 107-118, doi:10.1093/molbev/msx270 (2018).

2 Woods, S. et al. Duplication and Retention Biases of Essential and Non-Essential Genes Revealed by Systematic Knockdown Analyses. *PLoS Genet.* 9, doi:10.1371/journal.pgen.1003330 (2013).

3 He, X. & Zhang, J. Higher duplicability of less important genes in yeast genomes. *Mol Biol Evol* 23, 144-151 (2006).

4 Vance, Z., Niezabitowski, L., Hurst, L. D. & McLysaght, A. Evidence from *Drosophila* Supports Higher Duplicability of Faster Evolving Genes. *Genome Biology and Evolution* 14, doi:10.1093/gbe/evac003 (2022).

5 Nozawa, M., Suzuki, Y. & Nei, M. Reliabilities of identifying positive selection by the branch-site and the site-prediction methods. *Proc Natl Acad Sci U S A* 106, 6700-6705, doi:10.1073/pnas.0901855106 (2009).

6 Gharib, W. H. & Robinson-Rechavi, M. The branch-site test of positive selection is surprisingly robust but lacks power under synonymous substitution saturation and variation in GC. *Mol Biol Evol* 30, 1675-1686, doi:10.1093/molbev/mst062 (2013).

Point-by-point response to the reviewers' comments

Reviewers' comments are shown as **black** text. Response to comments is shown as **blue** text. All changes were indicated in the manuscript file by using yellow highlighting.

Reviewer(s)' Comments to Author:

Reviewer #1:

Yang et al., report on the large-scale discrepant region in the T2T compared to hg38. While this analysis has been performed previously, the authors identify some previously unidentified non-syntenic regions which is of utility to the genomics community. They deploy a novel web-based tool (Synplotter) to characterise these regions and identify KLRC as a cluster with potential functional relevance. While I do not have experience with the bioinformatic tools deployed here the analysis appears to be robust and well supported by orthogonal short-read and long-read data. The paper is clearly written and presented and unusually I did not find any faults with the paper. The novel insights are perhaps somewhat modest, but given the importance of accurate reference genomes and identifying structural differences between individual genomes I believe it merits publication.

Thank you for your time to review our manuscript. We extend our appreciation to you for the positive comments regarding the importance and strengths of our work.

Reviewer #2:

Yang et al. present a study comparing two important human genome assemblies: the current human genome reference (GRCh38) and the first telomere-to-telomere (T2T) human genome assembly (T2T-CHM13). The study identifies 590 discrepant regions between the two assemblies, which are highly structurally polymorphic in humans and likely associated with various human diseases. The study specifically focuses on a newly identified discrepant region, the KLRC gene cluster, which is associated with natural killer cell differentiation. And one gene KLRC3 has undergone rapid amino acid replacements during primate evolution. This study helps to further our understanding of the large-scale structural variation differences between these two crucial human reference genomes and their implications for the interpretation of studies of human genetic variation.

Thank you for your positive feedback on the significance of our work. Your thoughtful and constructive review is much appreciated.

The paper is clearly written and the results are solid given the application of variety of datasets as sort of cross validation. I only have several suggestions:

1. The genes identified in the discrepant regions seem from deletion events with respect to GRCh38, correct? Since only several genes were identified, it would be great to try to identify additional genes, if any, in other discrepant regions such as the insertion regions. A gene prediction tool could be applied or compare to some outgroups.

Thank you for the helpful suggestion to examine the gene differences in the discrepant regions between the two human reference genomes.

We revised our methods to more clearly define the structural types (Line 521):

"In this study, we defined structural types by detecting SVs on T2T-CHM13 with respect to GRCh38. Deletion refers a genomic region that is absent in T2T-CHM13 with respect to GRCh38, whereas SDR refers to a complex genomic difference rather than a simple deletion/insertion/inversion."

We then followed the reviewer's suggestion and used the RefSeq database and CAT (Comparative Annotation Toolkit) annotation tool to explore the gene differences in the

discrepant regions. We focused on the deletions and insertions, and identified 53 genes (including 22 protein coding genes) and 99 genes (including 21 protein coding genes) that differ in 28 deletions (28/68, 41.18%) and 43 insertions (43/87, 49.43%), respectively.

Rebuttal figure 1. The number of gene difference in deletions (total $N = 68$) and insertions (total $N = 87$) between GRCh38 and T2T-CHM13. Barplot shows the number of genes different (Y axis) in discrepant genomic regions between GRCh38 and T2T-CHM13. 28 deletion regions ($N = 28$, 28/68) and 43 insertion regions ($N = 43$, 43/87) include 53 genes (22 protein coding genes and 31 pseudogenes) and 99 genes (21 protein coding genes and 78 pseudogenes), respectively.

We revised our text in Line 168:

“In total, we examined the gene model difference in the discrepant regions, 22 and 21 protein-coding genes differ in the deletions and insertions, respectively (Additional file 1: Fig. S5, Additional file 2: Table S6).”

2. Because the discrepant regions are usually involved in complex genomic architectures (SD, CEN, TEL, TRF, etc.), it is challenging to genotype these regions. The application of read-depth genotyping from Illumina short reads data is potentially questionable - as reflected by the copy numbers range from 0 to over 10. The authors mentioned 94 HPRC long reads dataset. HGSVC also released 64 haplotype resolved genomes (Ebert et al. 2021). It would be great to genotype these regions comprehensively using these long reads. Based on the long-read results or pangenome results, PanGenie may be applied to accurately genotype these

regions.

Thank you for the constructive comment. While PanGenie is an excellent tool for genotyping SVs, it may not be suitable for detecting large-scale SVs [1]. As a result, we utilized the pangenome-graph of the 152 human long-read assemblies (88 HPRC + 64 HGSC) to directly examine their haplotypes [2].

It's worth noting that inversions have already been investigated in a previous study [3]. For this study, we chose to focus solely on insertions and deletions, we found that 66 and 39 deletions and insertions can be genotyped in the pangenome graph (Additional file 2: Table S7).

We revised our text in Line 187:

"We also used the long-read human pangenome graph (HPRC and HGSC, $n = 152$) to genotype the 87 insertions and 68 deletions [2, 4]. We found that 105 SVs (66 deletions and 39 insertions) can be genotyped in the pangenome graph (105/155, 67.74%) and the results suggest that the discrepant regions are polymorphic in humans (Additional file 2: Table S7)."

3. The authors identified approximately 226 Mbp discrepant regions. However, only the *KLRC* gene cluster looks significant and half of the manuscript is discussing this cluster, though very interesting. The molecular evolution analysis is very interesting. If the analysis can be applied to all the discrepant regions or at least the euchromatic regions, then it would definitely promote the work further. For example, the authors could simply check the divergent time along the non-human primate lineage for these regions. Are they human specific or common between species?

Thank you for the positive feedback regarding the evolutionary analysis of the *KLRC* gene cluster. We agree that a comprehensive analysis of the evolutionary history of discrepant regions would be valuable. As a first step, we utilized non-human primates (NHP), including bonobo, chimpanzee, gorilla, and orangutan, as outgroups to investigate the copy number of genes in the euchromatic regions. 39 protein-coding genes exhibited different copy number distributions between humans and nonhuman primates (Welch's t -test, Additional file 1: Table S14). Notably, 16 genes expanded in human lineage ("human-specific"), while the copy number of 23 genes varied among primates ("common between species").

However, we cannot perform the comprehensive evolutionary analyses of these genes in this study for two major reasons:

- (1) Some genes exhibit complex duplications/deletions during primate evolution, which would take several years to resolve (e.g., *NPIPs*).
- (2) Our colleagues are currently conducting research on certain genes (e.g., *GRPIN2*, *MUC3B*), and we do not wish to create conflicts with them.

Here, we examined the *GSTM* gene cluster reported in this study (Figure 2a), we firstly characterized the haplotypes of *GSTM* in NHP long-read genomes (Rebuttal Figure 2) and we extracted the genomic regions of *GSTM* to reconstruct the phylogenetic tree (Rebuttal Figure 3). Both the syntenic and phylogeny analyses show that *GSTM* genes recurrently mutated during the primate evolution.

Rebuttal figure 2. The syntenic relationship of *GSTM* gene cluster region between human and NHPs. The copy number *GSTM* is variable in primates and *GSTM2* is deleted in chimpanzee, bonobo, and gorilla genome assemblies.

Rebuttal figure 3. The phylogenetic tree of *GSTM* in primates. The phylogenetic tree of *GSTM* reconstructed with 4.1 kbp genomic regions suggests that *GSTM1* and *GSTM2* gene recurrently mutated during primate evolution.

We appreciate your understanding of our situation. Based on what we are able to present, we have revised our text in Line 423:

"The *GSTM* recurrently mutated during primate evolution with syntenic comparison and phylogenetic analyses (Additional file 1: Fig. S18-19, Additional file 2: Table S14)."

And we revised the text to discuss the limitation of this study regarding on the evolutionary analysis in Line 481:

"In addition, we explored the *KLRC* gene family in detail, but it was not possible to examine

the evolutionary history of each discrepant region in this study. Furthermore, we anticipate that we will gain a better understanding of the evolutionary history of each base in the human genome with the availability of complete primate genomes in the near future."

4. The GitHub page contains the visualization results of the 238 discrepant regions. It would be great to include all the scripts and codes of the results generated in the manuscript. This will largely increase the reproducibility of the work.

Thank you for the valuable suggestion. We uploaded all of the scripts used in our study to GitHub (https://github.com/YafeiMaoLab/discrepant_region.git). These include the scripts used for SV detection, CN polymorphism simulation, and evolutionary analysis.

Reviewer #3:

Yang and colleagues use the recent T2T human genome build to look for euchromatic regions that are new or discrepant from the standard GRCh38 build identifying 238 such regions, 67 of which were not previously recognised. They provide a useful tool to aid this. They also go beyond simple annotation as "non-syntenic" as done originally to characterize different modes of difference (insertions, deletions etc). They go on to examine gene models in the discrepant regions finding high copy number polymorphism rates. They also identify amino acid level differences in these genes suggesting polymorphism for both copy number and sequence. This motivates a call for investigation of these as possible polymorphisms associated with disease or unusual evolutionary patterns. They then move focus to consider a newly identified discrepant region gene *KLRC* identifying three haplotypes in humans (with 0,1, and 2 gene copies of *KLRC2*). Interestingly, the T2T genome has a *KLRC2* deletion. They identify duplication of *KLRC2* and *KLRC3* at the LCA of apes and old-world monkeys. They identify 3 amino acids in *KLRC3* that appear to have undergone positive selection. They identify a deletion in *KLRC2* in 20% of humans. This appears to be immune related (a known functionality associated with rapid evolution). From this they infer biological importance of the discrepant regions.

Thank you for carefully reviewing our manuscript and we appreciate the constructive comments.

This is a very impressive and sizeable body of research (and the above doesn't capture all the results!). Indeed, so sizeable that one must question whether this is in fact one paper. At one level it is very broad scale. At the end we consider three amino acids in one protein and details of protein structures and interactants. As is there doesn't seem to be an easy to grasp story here. This is rather reflected in the last line of the abstract that claims rather vaguely that the work "furthers our understanding...". Well, true it does, but it is revealing that nothing more specific can be claimed.

We are pleased that the referee thought the work is impressive in scale and while in principle multiple papers could be written. There are two major reasons why we focused on *KLRC2/3* gene family: (1) we intended to illustrate how these large-scale differences could affect the interpretation of a given gene within a gene family. (2) we intended to illustrate how the mixed haplotypes affect the association study (e.g., eQTL). Thus, this treatment, focusing on a

given gene, would broaden the appeal to the average evolutionary biologist reading this manuscript in *Genome Biology*. As referee suggested, more work of course needs to be done here but the signatures are sufficient to warrant an even deeper dive.

We discussed this issue in the Line 481:

" In addition, we explored the *KLRC* gene family in detail, but it was not possible to examine the evolutionary history of each discrepant region in this study. Furthermore, we anticipate that we will gain a better understanding of the evolutionary history of each base in the human genome with the availability of complete primate genomes in the near future."

Indeed, I would say I found this an especially difficult paper to review because of the changes in scale and absence of focus. Fortunately, I was asked to specifically consider the evolutionary aspects and so will restrict my comments to these issues. This being said, I was routinely confused as to whether the differences between the complete and incomplete builds are owing to incompleteness of the original build or bona fide variation between genomes in certain highly variable regions. If the latter then you don't need T2T you just need long reads and well built genomes. From what I can see the variation in the euchromatic sections seem to largely be the latter (suggested on line 158, CN polymorphism seems to be associated with discrepant regions) not the former. The interesting analysis of the misbuild of the *KLRC* region in GRCh38 was however also rather instructive.

This is a fair and constructive comment. In fact, the genomic differences are due to both incomplete builds (e.g., *GPRIN2*, *Nat Methods* (2022) [5]. Rebuttal figure 4a) and incompletely understood variation (e.g., *CRI*, Rebuttal figure 4b). Therefore, having the T2T build is crucial to assess both types of differences. Recent studies have shown that even with long-read genomes, there are still hundreds of gaps, most of which are associated with copy number polymorphic regions in human genomics [6]. While long-read genomes can resolve simple variable regions, they still face challenges in resolving complex 'hyper-variable' regions [6]. We appreciate the reviewer's input on this important topic.

Rebuttal figure 4. a The T2T genome revealed the presence of a new human-specific genomic region, *GPRIN2B*, which is located in a gapped region of GRCh38, as depicted in the top panel. The bottom panel illustrates a comparison of synteny among the genomes of humans, bonobos, and orangutans, which shows a region specific to humans containing the *SYT15*, *GPRIN2B*, and *NPY4R2* genes. This region would have been missed in comparative genomic studies without the T2T genome effort [5]. **b** An ~18.5 kbp deletion has resulted in the depletion of eight exons of *CR1* in T2T-CHM13, indicating a polymorphism in the *CR1* gene.

If the claim is that variable regions are interesting because they are variable then this needs much better framing. In the intro the authors argue that dynamic regions are important regions for both adaptation and disease (page 3 line 69). This strongly reflects old claims that after gene duplication duplicates evolve faster owing release from constraint and neo-functionalisation. However, more recently a more interesting possibility has been shown to be more likely: genes that change copy number and are fast evolving, are like this because they are unimportant. They always were fast evolving, their knockouts tend to have no phenotypes and they are possibly just neutral tolerated duplications 1-4. It would be helpful to frame the CNV and gene duplication in a manner that allows for these more recent possibilities. The idea that these variable regions may be neutral is touched on in the discussion, but not framed wrt this literature and evidence base either there or in the introduction.

We totally agree with the referee that we did not include other fates of duplicated genes in the manuscript. It is well established that duplicated genes and regions mutate an accelerated rate due to NAHR, inter-locus gene conversion and other mutational processes [7]. Recent work also shows that the duplicated genes are likely from 'non-essential'/'less important'/'smaller'/'faster evolving' genes based on yeasts (*Saccharomyces*), flies

(*Drosophila*), worms (*C. elegans*) and others species [8-11].

We reframed the Discussion to reflect these possibilities in Line 430:

"In addition, other studies on duplicated genes show that non-essential, less important, or fast-evolving genes are more likely to be duplicated during evolution [59-62]. Therefore, functional assessments of these polymorphic loci are worth considering in future studies."

For the sections I would like to comment on, and have looked at in some detail, I decided that I cannot really comment on the results as the methods are so poorly described that I don't know what has been done and assumed. Normal reporting standards have not been followed. Stress testing has not been performed. I think the general approach may be sound (I can't really say) but certainly not replicable. This is a common problem with expansive multi-story papers. I highlight:

Thank you for the constructive criticisms. We carefully addressed your comments point to point below.

* At the individual gene level evolution we are told that PAML was used with a branch and a branch-site model (Table S11). This is fine but in a paper on positive selection one would usually give many more details: what parameters were specified in each (codon model?), what degrees of freedom do you assume when comparing likelihoods of the nested models? What are the nested models? As is we are just presented with tree topologies, likelihoods and P values. Near uninterpretable I would suggest.

Thank you for pointing out the issue that we did not write more details of PAML selection tests in our previous manuscript.

We revised the methods in details in Line 678:

" In the branch model, we set the following parameters to establish the null model: 'runmode = 0, seqtype = 1, CodonFreq = 2 (F3X4), model = 2, NSsites = 0, getSE = 0, icode = 0, fix_kappa = 0, kappa = 1, fix_omega = 1 (omega fixed), and omega = 1'. We assumed that all branches have an omega value of 1, with $np = 13$ degrees of freedom. To test whether the *KLRC2* and *KLRC3* clades have different selection pressures (omega values), we utilized the same parameters as the null model, but with different user-specified trees and free dN/dS ratio

set for the two clades, with $np = 13$ as described in a previous study [12]. We observed that the *KLRC3* clade with the estimated omega model has a lower likelihood value ($p = 0.033$), indicating that the clade is not evolving neutrally.

In the branch-site model, we used the following parameters to set the null model: runmode = 0, seqtype = 1, CodonFreq = 2 (F3X4), model = 2, NSsites = 0, getSE = 0, icode = 0, fix_kappa = 0, kappa = 1, fix_omega = 0 (omega free), and omega = 1 (initial omega). This assumed that all branches have a free omega with $np = 14$ (degrees of freedom). To test for different selection pressures on different sites in the *KLRC3* clade, we used the following parameters: runmode = 0, seqtype = 1, CodonFreq = 2 (F3X4), model = 2 (user-specified dN/dS ratios for branches), NSsites = 2, getSE = 0, icode = 0, fix_kappa = 0, kappa = 1, fix_omega = 0 (omega free), and omega = 1 (initial omega), with $np = 16$ as described in a previous study. We observed that three amino acids have a possibility of being under positive selection greater than 0.9 in the *KLRC3* clade ($p = 0.006$), suggesting that these amino acids are likely under positive selection."

* The aBSREL analysis (line 300). As the aBSREL authors note there is an optimum complexity for BS models. What did you use and where are the results? Table S11 says it is just PAML. I'm confused.

Thank you for your constructive comment. As suggested by the referee, we explored the aBSREL model to test selection on a branch. In this study, we selected the clade node 2 (*KLRC3* clade) as the target for testing selection pressure using the full adaptive model in aBSREL (online tool).

summary
tree
table
model fits

Detailed results

Name	B	LRT	Test p-value	Uncorrected p-value	ω distribution over sites	
Node3	0.0327	62.7134	0.0000	0.0000	$\omega_1 = 0.00$ (95%) $\omega_2 = 5300$ (5.1%)	View
BONOBO_KLRC3	0.0078	test not run	1.0000	1.0000	$\omega_1 = 1.51$ (100%)	View
CHIMP_KLRC2	0.0162	test not run	1.0000	1.0000	$\omega_1 = 0.574$ (100%)	View
CHIMP_KLRC3	0.0148	test not run	1.0000	1.0000	$\omega_1 = 0.428$ (100%)	View
GORILLA_KLRC2	0.0044	test not run	1.0000	1.0000	$\omega_1 = 0.224$ (100%)	View
HUMAN_KLRC2	0.0120	test not run	1.0000	1.0000	$\omega_1 = 0.390$ (100%)	View
HUMAN_KLRC3	0.0092	test not run	1.0000	1.0000	$\omega_1 = 0.814$ (100%)	View
Node1	0.0072	test not run	1.0000	1.0000	$\omega_1 = 0.568$ (100%)	View
Node2	0.0163	test not run	1.0000	1.0000	$\omega_1 = 1.07$ (100%)	View
Node5	0.0101	test not run	1.0000	1.0000	$\omega_1 = 0.201$ (100%)	View
ORANGUTAN_KLRC2	0.0258	test not run	1.0000	1.0000	$\omega_1 = 0.00$ (86%) $\omega_2 = 6.80$ (14%)	View

Rebuttal figure 5. Selection pressure testing using aBSREL on *KLRC3* clade. The aBSREL model suggested that node 2 is under selection with a p -value of ≤ 0.05 .

We have included the details of these processes in the supplementary figure 20 and revised the method in line 665:

"Then, we ran a preliminary selection test on aBSREL

(<https://www.datamonkey.org/analyses>) and the aBSREL tool showed the selection signals on

the *KLRC3* clade. Specifically, we used the CDS alignment as input and selected the *KLRC3* clade branch for testing selection pressure with a full adaptive model. After a p-value correction, the aBSREL analysis with the full adaptive model revealed significant selection pressure in the *KLRC3* clade branch ($p = 0$, Additional file 1: Fig. S16)."

In total, both aBSREL and PAML branch models suggest the selection pressure on the *KLRC3* clade. The results support our statement in the manuscript.

* Fig 5a uses BEAST2 - input parameters, how run?

The divergence time was estimated using the HKY substitution model, relaxed lognormal clock model, and calibrated Yule prior with the divergence time reported previously.

We revised the method section in Line 644:

“Here, we used the log-normal and the real mean model to set the prior calibrate time, including pan-lineage split time (~1.45 mya), owl monkey and marmoset split time (~24.5 mya), monkey and ape split time (~54 mya) in this study. The divergence time was estimated using the HKY substitution model, relaxed lognormal clock model, and calibrated Yule prior with the divergence time described above. The MCMC chains were run 30,000,000 steps and 3,000,000 steps were set for burnin running. Finally, we used the tracer (v1.7.1) to examine whether the chain was convergent. Indeed, each ESS value of each parameter was over 200 in our study and these results suggested the MCMC chain was converged. We repeated this divergence time estimation three times independently, with each run converging and producing coincident estimated times. All results are available on our GitHub page (https://github.com/YafeiMaoLab/discrepant_region.git).”

* Are the PAML results robust to modifications? I note that the Branch models aren't overwhelmingly significant, but I also have no idea what you did. Branch-site models in particular are also rather notoriously prone to false positives, problems with saturation and GC content^{5,6} so stress testing the results is important.

Thank you for rising concerns regarding the PAML testing. As the referee suggested, the branch-site model is likely susceptible to false positives due to saturation and GC content [13-15]. We therefore examined the GC content and saturation in our case. Based on the

literatures, simulations showed that GC content between 35-65% has a high power to detect positive selection with a lower false negative rate [15]. The table below demonstrates that all the CDS sequences we used as input have standard GC content, and there is no significant variation of GC content between each sequence. Therefore, GC content should not affect our analysis in the branch-site model in this study.

Table1. GC content of *KLRC2&3* in different species.

Gene	GC content
Human_ KLRC3	41.8
Human_ KLRC2	40.4
Bonobo_ KLRC3	41.5
Chimpanzee_ KLRC3	41.4
Chimpanzee_ KLRC2	41.5
Gorilla_ KLRC2	40.8
Orangutan_ KLRC2	40.1

We also examined saturation in our study. According to the literature, saturation occurs when sequence divergence is too high, meaning the tree length is too large [14]. Studies showed that extremely high divergence (saturation) can cause a higher false positive rate. In our study, the tree length is equal to 0.52 (defined as the number of nucleotide substitutions per codon), and the average of *dN* and *dS* values for all branches are 0.0124 and 0.055, respectively. All empirical data in our case is not extremal data.

Additionally, Zhang et al. argued that sample size is a key factor that affects the robustness of the branch-site model. They propose a small sample model using Fisher's exact test. To test our empirical data with their model, we first estimated the nonsynonymous and synonymous sites on the *KLRC3* clade branch (see table below). We then used Fisher's exact test to examine our empirical data, and the test showed evidence of selection pressure ($p = 0.039$).

Table 2. The selection test on small sample size model

	Test of positive selection	
	Non.	Syn.
Changes	16	11
No Changes	534	159

We revised our text in Line 702:

"Our selection tests may be affected by several factors, including GC content, saturation, and small sample size [75-77]. To address these potential issues, we examined the GC content (GC: 40-42) and the saturation level (tree length: 0.52) in our empirical data, and the data fit for the model (Additional file 2: Table S15). We used Fisher's test with a small sample model to test for selection pressure (Additional file 2: Table S16), and the results were condicent with those obtained using PMAL."

* Fig S13 you claim no significant differences based on pi - line 298, method line 572. Differences between what and what? What test did you do? Second, you should control for local recombination rate (diversity is expected to be higher when recombination is lower owing to background selection).

Thank you for pointing out the unclear expression in our previous statement. To estimate the pi diversity in 94 long-read human assemblies, we utilized a window-sliding approach with a 20kb window and 10kbp sliding window, focused on the region (T2T-CHM13: chr12:10,000,000-10,700,000). Our analysis revealed that the pi diversity of the *KLRC2-KLRC3* region (T2T-CHM13: chr12:10,299,045-10,307,426) does not differ significantly from the entire regions (chr12:10,000,000-10,700,000).

Regarding the precise recombination rate, it could not be accurately inferred from the 94 long-read assemblies due to the small sample size ($n = 94$). Therefore, we used genome scan data directly from previous studies [16, 17], which included the recombination rate, to examine positive selection. These previous data show that there is no selection signal present in the region.

We revised this in Line 343 as follows: "The pi diversity analysis of the *KLRC* gene cluster based on 94 long-read genome assemblies showed no significant pi diversity drop in the *KLRC2-3* region (chr12:10,299,045-10,307,426) with respect to the entire regions (chr12:10,000,000-10,700,000) (Additional file 1: Fig. S15). Furthermore, considering a large human population dataset (human 1KG data) and recombination rate, previous selection scans do not detect any positive selection on the *KLRC2-KLRC3* region in human populations neither [16, 17]."

These are just some of the issues. In sum, this looks like a data/analysis rich paper with lots of potentially interesting results but I'm not sure it is one paper. Multiple much better described papers may be preferable (the KLRC story could be a standalone). As is, this is not replicable with results and methods more akin to late period Picasso than an exquisitely detailed Vermeer.

1 O'Toole, A. N., Hurst, L. D. & McLysaght, A. Faster Evolving Primate Genes Are More Likely to Duplicate. *Molecular Biology and Evolution* 35, 107-118, doi:10.1093/molbev/msx270 (2018).

2 Woods, S. et al. Duplication and Retention Biases of Essential and Non-Essential Genes Revealed by Systematic Knockdown Analyses. *PLoS Genet.* 9, doi:10.1371/journal.pgen.1003330 (2013).

3 He, X. & Zhang, J. Higher duplicability of less important genes in yeast genomes. *Mol Biol Evol* 23, 144-151 (2006).

4 Vance, Z., Niezabitowski, L., Hurst, L. D. & McLysaght, A. Evidence from *Drosophila* Supports Higher Duplicability of Faster Evolving Genes. *Genome Biology and Evolution* 14, doi:10.1093/gbe/evac003 (2022).

5 Nozawa, M., Suzuki, Y. & Nei, M. Reliabilities of identifying positive selection by the branch-site and the site-prediction methods. *Proc Natl Acad Sci U S A* 106, 6700-6705, doi:10.1073/pnas.0901855106 (2009).

6 Gharib, W. H. & Robinson-Rechavi, M. The branch-site test of positive selection is surprisingly robust but lacks power under synonymous substitution saturation and variation in GC. *Mol Biol Evol* 30, 1675-1686, doi:10.1093/molbev/mst062 (2013).

We thank the referee for the critical and insightful evaluation, particularly with regards to the evolutionary analysis. While both Picasso and Vermeer are nonetheless masterpieces but we have strived to make the paper more integrated and have introduced in the preamble the genome-wide analysis and the purpose of a detailed analysis so the *KLRC* gene family. All references (1-6) are included in the revised manuscript.

Finally, we appreciate the insightful and critical comments to improve our manuscript once again.

Reference:

1. Ebler J, Ebert P, Clarke WE, Rausch T, Audano PA, Houwaart T, Mao Y, Korbel JO, Eichler EE, Zody MC, et al: **Pangenome-based genome inference allows efficient and accurate genotyping across a wide spectrum of variant classes.** *Nat Genet* 2022, **54**:518-525.
2. Ebert P, Audano PA, Zhu Q, Rodriguez-Martin B, Porubsky D, Bonder MJ, Sulovari A, Ebler J, Zhou W, Serra Mari R: **Haplotype-resolved diverse human genomes and integrated analysis of structural variation.** *Science* 2021, **372**:eabf7117.
3. Porubsky D, Höps W, Ashraf H, Hsieh P, Rodriguez-Martin B, Yilmaz F, Ebler J, Hallast P, Maggiolini FAM, Harvey WT: **Recurrent inversion polymorphisms in humans associate with genetic instability and genomic disorders.** *Cell* 2022, **185**:1986-2005. e1926.
4. Wang T, Antonacci-Fulton L, Howe K, Lawson HA, Lucas JK, Phillippy AM, Popejoy AB, Asri M, Carson C, Chaisson MJ: **The Human Pangenome Project: a global resource to map genomic diversity.** *Nature* 2022, **604**:437-446.
5. Mao Y, Zhang G: **A complete, telomere-to-telomere human genome sequence presents new opportunities for evolutionary genomics.** *Nature Methods* 2022, **19**:635-638.
6. Porubsky D, Vollger MR, Harvey WT, Rozanski AN, Ebert P, Hickey G, Hasenfeld P, Sanders AD, Stober C, Korbel JO: **Gaps and complex structurally variant loci in phased genome assemblies.** *bioRxiv* 2022.
7. Vollger MR, DeWitt WS, Dishuck PC, Harvey WT, Guitart X, Goldberg ME, Rozanski A, Lucas J, Asri M, Consortium HPR: **Increased mutation rate and interlocus gene conversion within human segmental duplications.** *bioRxiv* 2022:2022.2007. 2006.498021.
8. Vance Z, Niezabitowski L, Hurst LD, McLysaght A: **Evidence from Drosophila Supports Higher Duplicability of Faster Evolving Genes.** *Genome Biol Evol* 2022, **14**.
9. Woods S, Coghlan A, Rivers D, Warnecke T, Jeffries SJ, Kwon T, Rogers A, Hurst LD, Ahringer J: **Duplication and retention biases of essential and non-essential genes revealed by systematic knockdown analyses.** *PLoS Genet* 2013, **9**:e1003330.
10. He X, Zhang J: **Higher duplicability of less important genes in yeast genomes.** *Mol Biol Evol* 2006, **23**:144-151.
11. O'Toole Á N, Hurst LD, McLysaght A: **Faster Evolving Primate Genes Are More Likely to Duplicate.** *Mol Biol Evol* 2018, **35**:107-118.
12. Cantsilieris S, Sunkin SM, Johnson ME, Anaclerio F, Huddleston J, Baker C, Dougherty ML, Underwood JG, Sulovari A, Hsieh P, et al: **An evolutionary driver of interspersed segmental duplications in primates.** *Genome Biol* 2020, **21**:202.
13. Zhang J, Kumar S, Nei M: **Small-sample tests of episodic adaptive evolution: a case study of primate lysozymes.** *Mol Biol Evol* 1997, **14**:1335-1338.
14. Nozawa M, Suzuki Y, Nei M: **Reliabilities of identifying positive selection by the branch-site and the site-prediction methods.** *Proc Natl Acad Sci U S A* 2009, **106**:6700-6705.
15. Gharib WH, Robinson-Rechavi M: **The branch-site test of positive selection is surprisingly robust but lacks power under synonymous substitution saturation and variation in GC.** *Mol Biol Evol* 2013, **30**:1675-1686.
16. Liu X, Ong RT, Pillai EN, Elzein AM, Small KS, Clark TG, Kwiatkowski DP, Teo YY: **Detecting and characterizing genomic signatures of positive selection in global populations.** *Am J Hum Genet* 2013, **92**:866-881.
17. Voight BF, Kudaravalli S, Wen X, Pritchard JK: **A map of recent positive selection in the human genome.** *PLoS Biol* 2006, **4**:e72.

Second round of review

Reviewer 2

The authors have addressed my concerns.

Reviewer 3

The authors have done a very good job in responding to my concerns: the details necessary for replication (or at least for comprehension) are given. The framing is better. Naturally, I am happy to respect the authors choice of what should be in the paper (and this is now somewhat better motivated).

I have no further concerns.

PS I realised I miswrote when I said diversity would be higher when recombination is lower - sorry for any confusion: it should be lower diversity with lower recombination owing the background selection.